# Severe COVID-19 patients have impaired plasmacytoid dendritic cell-mediated control of SARS-CoV-2

Manon Venet [1,3], Margarida Sa Ribeiro [1,3], Elodie Décembre [1,3], Alicia Bellomo[1,3], Garima Joshi[1], Célia Nuovo[1], Marine Villard [1], David Cluet [2], Magali Perret[1], Rémi Pescamona[1], Helena Paidassi [1], Thierry Walzer [1], Omran Allatif [1], Alexandre Belot [1], Sophie Trouillet-Assant[1], Emiliano P. Ricci [2] & Marlène Dreux [1] ✉

Type I and III interferons (IFN-I/λ) are important antiviral mediators against SARS-CoV-2 infection. Here, we demonstrate that plasmacytoid dendritic cells (pDC) are the predominant IFN-I/λ source following their sensing of SARS-CoV-2-infected cells. Mechanistically, this short-range sensing by pDCs requires sustained integrin-mediated cell adhesion with infected cells. In turn, pDCs restrict viral spread by an IFN-I/λ response directed toward SARS-CoV-2-infected cells. This specialized function enables pDCs to efficiently turn-off viral replication, likely via a local response at the contact site with infected cells. By exploring the pDC response in SARS-CoV-2 patients, we further demonstrate that pDC responsiveness inversely correlates with the severity of the disease. The pDC response is particularly impaired in severe COVID-19 patients. Overall, we propose that pDC activation is essential to control SARS-CoV-2-infection. Failure to develop this response could be important to understand severe cases of COVID-19.

The innate immune system acts as the first line of defense for the sensing of viral infection. This involves rapid recognition of pathogen-associated molecular patterns (PAMPs), including viral nucleic acids, by pattern recognition receptors (PRRs). This recognition results in an antiviral response characterized by the production of type I and III (λ) interferons (IFN) and other pro-inflammatory cytokines, along with the expression of IFN-stimulated genes (ISGs). Whilst type I and III/λ IFNs interact with distinct receptors, they both induce similar signaling pathways and effector factors, thus referred herein to as the IFN-I/λ response[1,2]. This host response suppresses viral spread by blocking the viral life cycle at multiple levels, hereby promoting virus clearance. The IFN-I/λ response also mediates immunomodulatory effects in surrounding tissues and imparts the onset of the adaptive immune response[3,4].

Severe acute respiratory syndrome coronavirus-2 (SARS-CoV-2) emerged in December 2019 and is responsible for the still-ongoing coronavirus disease 2019 pandemic[5]. Although most SARS-CoV-2-infected individuals experience asymptomatic to mild disease, others develop respiratory distress syndrome that is lethal in the most severe cases. Importantly, the IFN-I/λ response is now thought to be a critical host response against SARS-CoV-2 infection and its pathogenesis[2,6–10]. Recent studies have shown that SARS-CoV-2 can evade the initial control by the IFN-I/λ response via manifold inhibitory mechanisms interfering with both the sensing and the signaling pathways within infected cells (review articles[2,6,11,12] and illustrated in previous works[10,13–17]). This immune evasion might lead to an increased viral load, followed by widespread inflammation[6]. IFN-I/λ response is therefore thought as pivotal for the host defense against respiratory

[1]CIRI, Inserm, U1111, Université Claude Bernard Lyon 1, CNRS, UMR5308, École Normale Supérieure de Lyon, Univ Lyon, F-69007 Lyon, France. [2]Laboratory of Biology and Modeling of the Cell, Université de Lyon, ENS de Lyon, Université Claude Bernard, CNRS UMR 5239, Inserm, U1293 Lyon, France. [3]These authors contributed equally: Manon Venet, Margarida Sa Ribeiro, Elodie Décembre, Alicia Bellomo. ✉e-mail: marlene.dreux@ens-lyon.fr

infections. Individual immune responses against viral infection can thus be extremely heterogenous, ranging from robust and fast IFN-I/λ production to impaired IFN-I/λ-mediated immunity. Timing of the IFN-I/λ production is also a critical parameter in related Coronaviruses (e.g., MERS-CoV and SARS-CoV-1)[18]. Delayed IFN-I/λ signaling is associated with robust virus replication and promotes the accumulation of pathological monocyte-macrophages[18]. This results in lung immuno-pathology, vascular leakage, and suboptimal T-cell responses. Along this line, early reports on SARS-CoV-2 suggest that severe COVID-19 featured low level of IFN-I/λ but overproduction of inflammatory cytokines[7,19–21]. Accordingly, genetic deficiency (e.g., X-linked recessive TLR7 deficiency), neutralization by autoantibodies directed against the IFN−I system, or viral-mediated inhibition of the IFN-I/λ response aggravates SARS-CoV-2 pathogenesis[21–27]. It is therefore critical to understand the regulation of the optimal production and activity of IFN-I/λ.

The plasmacytoid dendritic cells (pDC) are a unique immune cell type specialized for rapid and massive production of IFN-I/λ[28]. pDCs possess multiple adaptations to efficiently produce IFN-I/λ[28–32]. As pDCs are refractory to most viral infections, their response is not directly repressed by viral proteins. This particularity contributes to the exceptional magnitude of pDC IFN-I/λ production[28,33]. The main viral-sensors responsible for pDC activation are Toll-like receptors (TLR)-7 and −9 that recognize viral RNA and DNA, respectively[28]. Recent reports suggested that pDCs activated by SARS-CoV-2 differentiate into cytokine- and IFN-secreting effector cells, in a TLR-dependent-manner[34–36]. Studies on related coronaviruses have demonstrated that pDCs migrate into the lungs upon infection[37], and others have also demonstrated a viral control by pDC-derived IFN-I in this context[38]. Reports on SARS-CoV-2 showed that an early and transient IFN-I response is associated with moderate COVID-19 disease (e.g.[19]). Nonetheless, how pDCs respond to SARS-CoV-2-infected cells and how this response correlates with the progression of COVID-19 severity are still open questions.

Here, we explore the molecular mechanisms underlying the IFN-I/λ response against SARS-CoV-2 infection. Our results indicate that pDCs establish cell contact with SARS-CoV-2 infected cells via $\alpha_L\beta_2$ integrin/ICAM-1 adhesion complex and regulators of the actin network. This physical contact between pDCs and infected cells is required for the pDC-mediated antiviral response by TLR7 recognition. Capitalizing on our findings that pDCs strongly respond by physical sensing of SARS-CoV-2-infected cells, we then show that impaired pDC IFN-I/λ response associates with COVID-19 severity. It is now increasingly recognized that pDCs differentiate into different subsets with distinct phenotypes and functionalities. Here, we showed that the differentiation of pDCs into subsets is altered when stimulated by contact with SARS-CoV-2-infected cells as compared to other activation signals. Especially, when in contact with SARS-CoV-2-infected cells, pDCs preferentially differentiate into a subset population that efficiently produces IFN-I/λ, leading to a robust antiviral control directed towards the infected cells.

## Results

### Robust activation of pDCs in response to SARS-CoV-2-infected cells

Respiratory epithelial cells represent the first infected tissue in the course of SARS-CoV-2 infection. To investigate which hematopoietic cell type is primarily responsible for the IFN-I/λ response against SARS-CoV-2 infection, human peripheral blood mononuclear cells (PBMC) isolated from healthy donors were cocultured with SARS-CoV-2-infected human lung-derived cells. Calu-3 cells and A549-ACE2 cells (expressing the angiotensin-converting enzyme 2) were infected for 48 h prior to coculture with PBMCs [tPBMC] for 16−18 h. We found that PBMCs respond by a robust secretion of IFNα when cocultured with SARS-CoV-2-infected cells (Fig. 1a, b, [coculture]). In contrast, the

supernatants [SN] of SARS-CoV-2-infected cells failed to trigger IFNα secretion by PBMCs (Fig. 1a, b). As expected from previous publications, SARS-CoV-2-infected cells did not produce detectable levels of IFNα[19,39–44]. Plasmacytoid dendritic cells (pDCs) are known to robustly produce IFN-I/λ, notably IFNα[28]. In line with this, antibody-mediated depletion of pDCs from PBMCs abolished IFNα secretion in response to cocultured SARS-CoV-2-infected cells (Fig. 1a, b). To further demonstrate that pDCs are the major source of IFNα upon incubation with SARS-CoV-2-infected cells, pDCs were purified from PBMCs (Supplementary Fig. 1a). Purified pDCs potently produced IFNα in response to coculture with SARS-CoV-2-infected cells (Fig. 1a, b, [iso pDCs]).

Consistent with these results, we further showed that IFNα producer cells were markedly enriched in the cell population gated as pDCs as compared to other hematopoietic cell types (i.e., non-pDCs): no IFNα⁺ cells detected among non-pDCs (Fig. 1c, d). We further demonstrated that other DC subsets, referred to as non-pDC enriched mDCs, and further gated as mDC1 and mDC2 did not produce detectable IFN-I/λ upon contact with infected cells (Fig. 1c, d, and see gating strategy in Supplementary Fig. 1a). Most IFNα⁺ pDCs concomitantly produced IFN-λ in response to SARS-CoV-2-infected cells (Fig. 1c, d). In contrast, the stimulation by soluble TLR agonists (R848 and polyI:C) elicited to some extend IFN-λ⁺ pDCs, but no detectable IFNα⁺ cells (Fig. 1c, d and Supplementary Fig. 1b), yet a potent upregulation of surface expression of activation markers including CD83 and the programmed cell-death ligand-1 (PD-L1) as compared to coculture with SARS-CoV-2-infected cells (Supplementary Fig. 1c, d). This suggested that the pDC response to SARS-CoV-2-infected cells is likely qualitatively distinct from stimulation by soluble agonist. CD83 and PD-L1 are induced by NF-κB-signaling, and thus dependent on signaling distinct from the IFN-I/λ response[45,46].

Next, we tested whether the sensing of SARS-CoV-2-infected cells required a productive infection of pDCs, using a recombinant SARS-CoV-2 infectious clone expressing the mNeongreen reporter [icSARS-CoV-2-mNG][47]. No infection of pDCs (defined as CTV⁺-pDCs) was detected when pDCs were incubated in contact with icSARS-CoV-2-mNG-infected A549-ACE2 cells (Supplementary Fig. 1e, upper panels). However, in the same coculture set up, infection by icSARS-CoV-2-mNG was readily detected in initially uninfected/naive RFP⁺ A549-ACE2 cells (Supplementary Fig. 1e, lower panels), hence proving validation of efficient viral transmission.

Taken together, these results strongly suggest that IFNα is robustly produced only by pDCs. This occurs by sensing of SARS-CoV-2-infected cells without productive infection and likely induces a specific activation state in pDCs.

### Short-range sensing of infected cells via cell-contact triggers the pDC IFN-α response

We observed that cell-free SN from SARS-CoV-2-infected cell types failed to trigger IFNα production by PBMCs or by purified pDCs (Fig. 1a, b), even using a multiplicity of infection (MOI) of 5 per pDC. Of note, to avoid possible misinterpretation due to contamination by cell debris and/or floating cells, we selected here an incubation time for SARS-CoV-2 infection so that no cytolytic effect was detectable in infected human lung-derived cells when cells/SNs were collected for coculture. This specific set up might contribute to the different observation compared to previous reports showing a pDC IFNα production triggered by SARS-CoV-2 SN[34–36]. Importantly, to further determine if cell-to-cell contacts were required for the transmission of the immunostimulatory signal to pDCs, we used transwell chambers containing SARS-CoV-2-infected cells and pDCs separated by a 0.4 μm permeable membrane. This physical cell separation fully prevented IFN-α production by pDCs (Fig. 1e, f). To confirm that this feature was not cell type-specific, we used a variety of lung-derived cell lines Calu-3, A549-ACE2, H358-ACE2 as well as non-lung Huh7.5.1 cells. We found

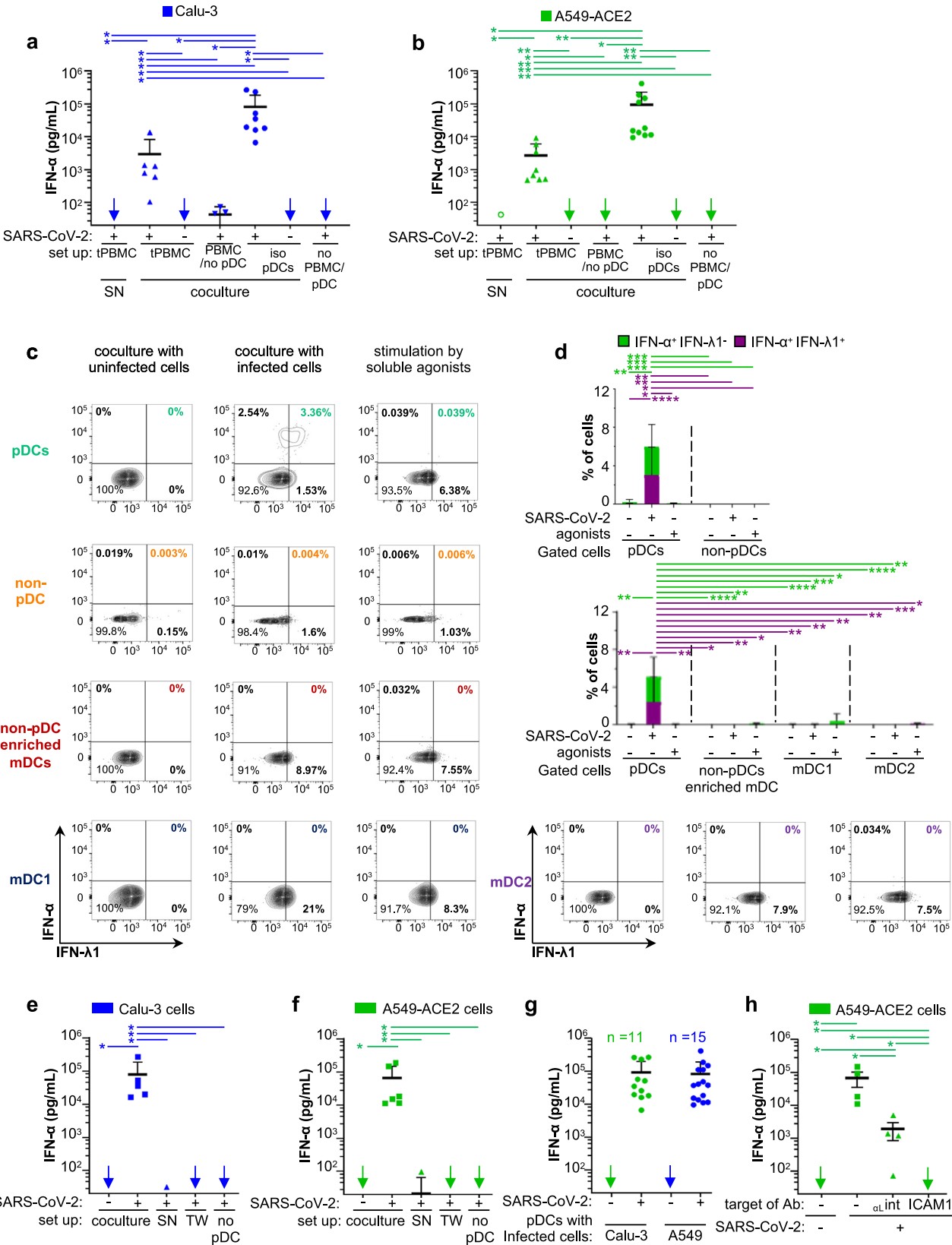

that pDCs were not stimulated by cell-free virus produced by any of these cell types. In accordance, pDCs were activated by all tested SARS-CoV-2-infected cells—when in direct contact - but not upon physical separation (Fig. 1e, f and Supplementary Fig. 1f, g). As control, we validated that pDCs were induced by agonist stimulation in this device (Supplementary Fig. 1h). Remarkably, similar levels of IFNα secretion

were reproducibly obtained for pDCs isolated from several distinct healthy donors (Fig. 1g; as $n = 11$ and $n = 15$ healthy donors for A549-ACE2 and Calu-3 cells, respectively).

To further define the mechanisms underlying pDC activation upon contact with SARS-CoV-2-infected cells, we assessed the implication of cell adhesion complexes in this process. We focused

**Fig. 1 | pDCs are the main IFN-α producers in response to contact with SARS-CoV-2-infected cells.** Cells were infected by SARS-CoV-2 (indicated as +) or not (indicated as −) 2 days prior to coculture with PBMCs, pDC-depleted PBMCs, and isolated pDCs (16 h-coculture). Infected cell types included the human alveolar basal epithelial cell lines, i.e., Calu-3 and A549-ACE2. Statistical analyses of the results were performed using Wilcoxon rank-sum test and p values were calculated with Tukey and Kramer test. For the different panels, the significant contrasts are indicated when p-values are: ≤0.05 as *; ≤0.005 as **; ≤0.0005 as ***; and ≤0.00005 as ****. **a, b** Quantification of IFN-α in the supernatants of total PBMCs [tPBMC], pDC-depleted PBMCs [PBMC/no pDC], and isolated pDCs [iso pDC] cocultured with either SARS-CoV-2-infected or uninfected Calu-3 cells (**a**) or A549-ACE2 cells (**b**), or treated with 100 μl of cell-free supernatant (SN) collected from SARS-CoV-2-infected cells. Of note, viral titers of SARS-CoV-2: SN ≈ 2.5 × 10⁵ foci forming units (ffu)/ml, MOI ≈ 1 per pDCs and no detection of IFN-α by SARS-CoV-2-infected cells themselves [no PBMC/pDC]. Arrows indicate results below the detection threshold of the IFN-α ELISA (i.e., 12.5 pg/ml). Each dot represents one independent experiment performed with distinct healthy donors (ELISA results are similarly presented in all other Figures). Error bars represent the means ± standard deviation (SD); from independent experiments (n = 5, 8, and 9 for the conditions of cocultures, respectively, with PBMCs, pDCs/Calu-3 cells, and pDCs/A549-ACE2 cells); exact p-values are indicated to Supplementary Fig. 8. **c, d** Total PBMCs were cocultured with SARS-CoV-2-infected or uninfected A549-ACE2 cells or treated with TLR agonists [31.8 μM R848 and 42.22 μM polyI:C] for 14–16 h. Cell populations gated as pDCs, non-pDC PBMCs, non-pDC enriched mDCs, mDC1, and mDC2 subsets, see gating strategies in Supplementary Fig. 1a. Representative dot blots of flow cytometry analyses (**c**) and frequencies of cells (**d**) positive for IFN-α⁺ but IFN-λ1⁻ or double positive for IFN-α⁺/IFN-λ1⁺ in gated pDCs versus non-pDC PBMCs (upper panels) and in gated pDCs versus non-pDC enriched mDCs, mDC1 and mDC2 subsets (lower panels). Means ± SD; Bars represent n = 10–11 independent experiments/distinct healthy donors. **e–h** Quantification of IFN-α in SNs of pDCs cocultured with the indicated cell types infected or not by SARS-CoV-2. **e, f** pDCs were cocultured with infected cells, either in direct contact [coculture] or physically separated by the semi-permeable membrane of transwell [TW], or treated with SN from the corresponding SARS-CoV-2-infected cells. IFN-α concentration was also determined in the SN of SARS-CoV-2-infected cells cultured without pDC (no pDC). Means ± SD; n = 5 independent experiments for pDC cocultured with infected cells or SN and n = 4 for TW and no pDC. **g** Dots represent IFN levels for pDCs purified from distinct donors in independent experiments; including n = 11 and n = 15 for A549-ACE2 and Calu-3 cells, respectively. Means ± SD. **h** Quantification of IFN-α in SNs of pDCs cocultured with SARS-CoV-2-infected cells A549-ACE2 treated or not with blocking antibodies against α_L-integrin and ICAM-1 at 10 μg/mL; means ± SD; each dots represent n = 4 independent experiments. Source data are provided as a Source Data file.

on α_L integrin and its ligand intercellular adhesion molecule (ICAM)-1 (also called CD54)[48], respectively, highly expressed by pDCs and by various cell types, guided by previous studies on the regulation of pDCs in the context of other infections[49]. Antibody-mediated blockade of both α_L integrin and ICAM-1 greatly prevented pDC IFN-α production (Fig. 1h and Supplementary Fig. 1i). The engagement of integrins by their ligands is known to induce local recruitment of the actin network. Notably Arp2/3 complex mediates actin nucleation by recruiting and branching actin filaments within the network[50–53]. Pharmacological inhibition of Arp2/3 complex impaired IFN-α production by pDCs in coculture with SARS-CoV-2-infected cells, in a dose dependent-manner (Supplementary Fig. 1j). This suggested that cell adhesion-induced actin recruitment is likely involved in the structuration of cell contacts. Furthermore, using specific TLR7 antagonist (i.e., IRS661), we showed that the endosome-localized TLR7 sensor mediates the sensing of SARS-CoV-2-infected cells by pDCs (Supplementary Fig. 1k).

Together our results demonstrated that physical contact between pDCs and SARS-CoV-2-infected cells is required for pDC IFN-α production. This contact involves cell adhesion mediated by α_L integrin/ICAM-1 complexes, which likely remobilize the actin network at the cell-to-cell contact, and leads to a robust TLR7-dependent IFN-α response by the pDCs.

## IFN-I/λ signature in patients at early time-point of SARS-CoV-2 infection

Based on the findings that pDCs are the main cell type producing IFNα in response to SARS-CoV-2-infected cells, we sought to explore how this singular activation mechanism for IFN-I/λ response is modulated in the course of the infection in patients and how it could relate to COVID-19 severity. A longitudinal study of IFN-I/λ response was done with different subsets of patients: (i) critically ill, herein referred to as severe patients, who presented acute respiratory distress syndrome or severe pneumonia at hospital admission and required mechanical ventilation in intensive care units, and (ii) patients with mild symptoms (i.e., low-grade fever, cough, malaise, rhinorrhea, sore throat), that group was sub-divided according to the days of sample collection post-symptom onset, i.e., mild/asymptomatic early for the first two weeks and mild/asymptomatic late at the later time points. Of note, most patients were SARS-CoV-2 positive in nasal swab samples by qPCR at sampling time in the mild/asymptomatic early group but not anymore or presenting low viral levels in the mild/asymptomatic late group. All patients and the analyzed time-points are listed in Table 1 and Supplementary Table 1, which provides information on the clinic and viral loads in nasal swab samples.

We quantified the IFN-I/λ levels in blood samples of infected patients at both transcriptional and protein levels for secreted IFN-α, IFN-λ1, IFN-γ, and IL-6. Both approaches showed an elevated IFN-I/λ response at early time points (within the first 10–11 days post-onset of symptoms) for all patients with mild symptoms/asymptomatic, that seemed to vanish over time, mirroring the controlled decrease of the viral load (Fig. 2a and Supplementary Table 1) and IFN-α were undetectable by day 40 post-symptom for all the patients of mild/asymptomatic group (Fig. 2a, second panel). As opposed, IFN-I/λ response was elevated in some severe COVID-19 patients at late time-points. Of note, the pro-inflammatory cytokine IL-6 was greatly detected in severe patients as compared to other groups (Fig. 2a and

## Table 1 | Demographical and clinical characteristics of health care workers (HCWs) with mild/asymptomatic and severe/critically ill COVID-19 patients

|  | Mild/asymptomatic COVID-19 n = 6 | Severe/critically-ill COVID-19 n = 6ᵃ |
|---|---|---|
| Demographics |  |  |
| Age, median [IQR] | 40.5 [24–57] | 63.5 [48.0–73.0] |
| Sex |  |  |
| Male, n (%) | 2 (33) | 4 (67) |
| Female, n (%) | 4 (67) | 2 (33) |
| BMI (kg/m²) | 23 [17–27] | 35.5 [28–49] |
| Clinical characteristics |  |  |
| Delay between symptom onset and ICU admission, day median [IQR] | N/A | 8 [3–26] |
| ICU length of stay, day median (min–max) | N/A | 45 [23–92] |
| Mortality, n (%) | 0 (0) | 4 (67) |
| Auto-ab anti-IFN-I, n (%) | 0 (0) | 1 (17) |

All laboratory data were recorded at recipient inclusion. *BMI* body mass index, *IQR* interquartile, *N/A* not applicable, *ICU* intensive care unit.
ᵃone of the severe/critically-ill COVID-19 patients was detected as positive for IFN-α.
Detection using an anti-IFN-α antibody - Human ELISA Kit (Thermo-fisher; ref. BMS217)
Results >1000 ng/mL (detection in non-positive individual is <34 ng/mL)
Detection >1000 ng/mL associates with neutralizing antibody.

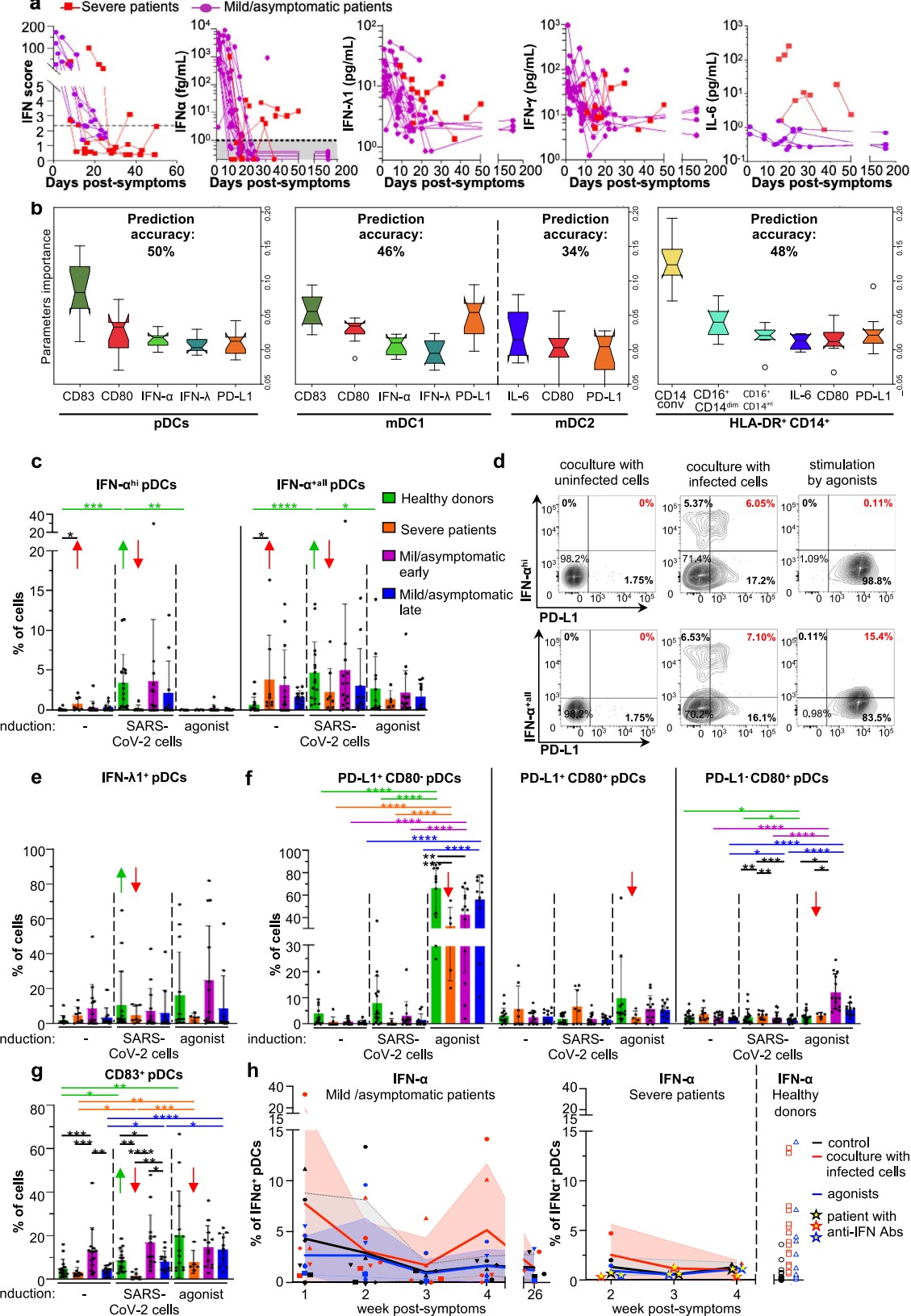

Supplementary Table 1). This positive correlation of the inflammatory response with COVID-19 severity is in agreement with previous reports[7,54–56].

Collectively, our results demonstrated distinct cytokine profiles across COVID-19 severities, and allow us to assign patient groups and insights on the associated biomarkers.

## COVID-19 severity correlates with a blunted pDC IFN-α response to SARS-CoV-2-infected cells

We then determined the ability of PBMCs isolated from the different groups of patients to respond to ex-vivo stimulation by SARS-CoV-2-infected cells. As our results pointed out the importance of pDCs in the IFN-I/λ response mounted against SARS-CoV-2-infected cells, we

**Fig. 2 | Innate responses against SARS-CoV-2-infected analyzed for patients with various COVID severities. a** Kinetic analysis of cytokine profile of severe and mild/asymptomatic COVID-19 patients. Determination of the IFN score in whole blood as well as secreted IFN-α, IFN-λ1, and IFN-γ, and IL-6 in plasma, at the indicated time-points. **b**–**i** PBMCs issued from the indicated groups of patients (i.e., severe, mild/asymptomatic early, mild/asymptomatic late, and healthy donors, see description of the patients in Table 1 and Supplementary Table 1, and Material and Methods section) were cocultured for 14–16 h with SARS-CoV-2-infected or uninfected A549-ACE2 cells, or treated with agonists [31.8 μM R848 and 42.22 μM polyI:C or 2.04 μM LPS], followed by the multiparametric analysis using flow cytometry. **b** Analysis of the flow cytometry dataset using a Machine learning approach based on Gradient Boosting. Upper-titles indicate the predictive accuracy in the validation set of samples for each cell type gated as pDCs, mDC1, and mDC2 subsets and HLA-DR⁺CD14⁺ monocytes (see gating strategies in Supplementary Fig. 2a). Graphs display the importance of the parameters in predicting the severity/group of patients from the validation set, via comparative analyses of all cell surface expressed-differentiation markers and intracellular cytokines defined in different cell types. Error bars correspond to the standard deviation of the mean importance of each parameter for each of the 10 downsampling iterations over all iterations. The bounds of the box plots correspond to the Interquartile Range (IQR) and the median is displayed as a line in the box. Notches represent the confidence interval (CI) around the median. In case values of the CI are less than the lower quartile or greater than the upper quartile, the notches will extend beyond the box, giving it a distinctive <<flipped>> appearance. The lower whisker corresponds to [Q1–1.5 × IQR (where Q1 corresponds to the first quartile)], while the upper whisker corresponds to [Q3 + 1.5 × IQR (where Q3 corresponds to the third quartile)]. Beyond the whiskers, data are considered outliers and are plotted as individual points. **c**, **d** Quantification of the frequency of cells positive for IFN-αʰⁱ/IFN-α⁺ᵃˡˡ (**c**) and representative dot blots of flow cytometry analyses (**d**). **e**–**g** Quantification of the frequency of IFN-λ1⁺ (**e**) and the differentiation markers CD80, PD-L1 (**f**) and CD83 (**g**) in gated pDCs. **c**, **e**–**g** Error bars represent the means ± SD; each dot represents the level determined for PBMCs from one individual patient in each group, or healthy donors and for all experimental condition; the *p* values are indicated as follows: ≤0.05 as *; ≤0.005 as **; ≤0.0005 as ***; and ≤0.00005 as ****. **h** Kinetic analysis of the ability of pDCs from patients referred to as mild/asymptomatic (left panel), severe (middle panel), and healthy donors (right panel) to mount an IFN-α⁺ response upon ex vivo stimulation with SARS-CoV-2-infected cells (red) agonist (blue) versus control cells (grey). Dots for the severe COVID-19 patient with circulating anti-IFNAR antibodies (see description in Supplementary Table 1) are represented by yellow-centered stars. Patient PBMCs were collected from the symptom onset and results correspond to the timeframes as follows, defined in weeks: 1 = [Days 1–8]; 2 = [Days 8–15]; 3 = [Days 15–22]; 4 = [Days 22–30]. Means (colored lines) and errors (colored areas) are indicated (*n* = 20–24 analyzed patients). Source data are provided as a Source Data file.

primarily analyzed pDCs. This response was compared to the one induced by TLR7 [R848]/TLR3[polyI:C] agonist. A group of *healthy* donors comprising similarly treated samples was used as reference. Post-stimulation, we performed a multiparametric flow cytometry analysis to define the profiles of cell surface expression of activation markers and intracellular cytokines. These expression profiles were further assigned to different pivotal innate immune cells by designing a panel of cell-type markers (see the gating strategy in Supplementary Fig. 2a).

A gradient boosting machine learning method was implemented using the flow cytometry datasets including all cell types (i.e., pDCs, HLA-DR⁺CD14⁺, mDC1, and mDC2) and parameters (i.e., IFN-α, IFN-λ, CD80, CD83, PD-L1, IL-6, CD14, CD16) analyzed. For each cell type, patient samples were divided into training/test (80%) and validation (20%) datasets. This set was then used to build a model using cross-validation with 80% of the training set for building and 20% to test, in which all parameters/markers of the studied cell type were used as variables for the predictors. To monitor any bias in the choice of samples for the training and validation sets, the split process was performed 10 times independently. Moreover, the imbalance between the three classes (healthy, mild, severe) was taken into account by performing 10 downsampling for each dataset. A model was then generated for each downsampling. The obtained models were then used to predict the patient status from the validation sets (i.e., using patient samples excluded from the model-building step). By combining different markers and stimulations of cell types relevant to the disease, Our models indicate that pDCs are the best predictors of patient status in the validation set (50% prediction accuracy), followed by HLA-DR⁺CD14⁺ and mDC1 cells (with a 48 and 46% prediction accuracy, respectively) (Fig. 2b). Expression markers from mDC2 had no significant predictive value, as the accuracy of the model on the validation set (34%) was close to the random assignment (33%) of patients between the healthy, mild and severe groups (Fig. 2b). Among the best predictive parameters within pDCs, we identified CD83 and CD80, followed by IFN-α (Fig. 2b). Among mDC1, CD83 and PD-L1 were the best predictors of patient status, while CD14⁺ conventional cells (i.e., CD14⁺ CD16⁻, see the gating strategy in Supplementary Fig. 2a) was the best predictive parameter for HLA-DR⁺ CD14⁺ cells. Of note, a distinction for IFN-αʰⁱ and IFN-αᵃˡˡ⁺ cells led to similar predictive accuracy (Supplementary Fig. 7e).

We further analyzed the expression of activation markers and cytokines individually at the single-cell level, focusing first on markers identified as driving forces in our machine-learning analyses. Similar to Fig. 1, the frequency of IFN-α producer pDCs greatly increased in response to SARS-CoV-2-infected cells in the *healthy* donor group for levels of both IFN-α⁺ᵃˡˡ and IFN-αʰⁱ (Fig. 2c, d, green arrows). The detection IFN-α was further validated by controls, including isotype control and omission of only anti-IFN-α within the same panel of antibodies i.e., keeping all the other antibodies of the staining panel, and using similar Flow cytometry settings (Supplementary Fig. 2b). The IFN-αʰⁱ pDCs correspond to the gating of pDCs highly positive for IFN-α (Fig. 2d). In sharp contrast, pDCs from severe COVID-19 patients failed to be activated by SARS-CoV-2-infected cells, as demonstrated by the absence or low detection of IFN-α, IFN-λ, CD83, and CD80/PD-L1 as compared to healthy donors and mild/asymptomatic patients (Fig. 2c–h; red bars and arrows). Of note, a strong basal level of pDC IFN-α expression was observed in the absence of ex-vivo stimulation for the severe COVID-19 patients (using less stringent discriminative gating of positive pDCs, noted distinctively as to IFN-α⁺ᵃˡˡ; Fig. 2c in red; right panel). The response to agonist stimulation was also greatly limited in pDCs from severe COVID-19 patients compared to healthy donors and patients with mild symptoms/asymptomatic, notably as shown by the level of activation markers (CD83, CD80, and PD-L1; Fig. 2f, g; red arrows). In accordance with the identified important predictive parameters using gradient boosting machine learning methods, the major change in pDC responsiveness in severe COVID-19 patients compared to the other groups was primarily explained by lack of IFN-α and CD83 expression in pDCs upon stimulation by SARS-COV-2-infected cells.

In addition to pDCs, other cell types were also selected for multiparametric flow cytometry analysis based on their potential to play a pivotal first defense against viruses. Especially, the Th1-promoting myeloid/conventional dendritic cell subset (mDC1), which produces IFN-λ via TLR3-mediated recognition of viral RNA and the Th2-promoting myeloid/conventional DC subset (mDC2), and monocytes i.e., Human Leukocyte Antigen−DR isotype (HLA-DR)⁺CD14⁺ populations, known to produce pro-inflammatory cytokines[57,58]. In line with the results obtained with pDCs, expression of IFN-λ1 was barely detectable in the mDC1 subset of severe COVID-19 patients upon stimulation by SARS-CoV-2-infected cells and agonists, comparatively to the other groups (Supplementary Fig. 2c; red arrows). As expected, in other cell populations (i.e., mDC1, mDC2, non-mDC2 and HLA-DR⁺CD14⁺ populations), other markers (i.e., IFN-α, IL-6, CD83, CD80, and PD-L1) were not readily induced by SARS-CoV-2-infected cells even in healthy donors (Supplementary Fig. 2d–g). Nevertheless, our data showed a potent upregulation of IL-6 by HLA-DR⁺CD14⁺ subset upon agonist treatment in

severe and mild/asymptomatic early group of patients (Supplementary Fig. 2d). HLA-DR⁺CD14⁺ population represents a highly frequent cell subset of non-mDC2 (e.g., among the gated live cells⁺ lin⁻ HLA-DR⁺: 65.7% and with exclusion of the few mDC2; Supplementary Fig. 2a). In accordance, a high frequency of IL-6⁺ cells was also observed for the non-mDC2 populations (Supplementary Fig. 2d, right panel). Likewise, severe COVID-19 patients presented an elevated level of blood IL-6 (Fig. 2a). These results are consistent with the previously reported production of pro-inflammatory cytokines by monocytic cells in patients with severe disease[7,55], and provided further insights into the differential responses of other immune cells.

### Dynamics of the IFN-I/λ response in COVID-19 patients

Next, we sought to define the dynamics of the response in COVID-19 patients. First, kinetic analyses in mild/asymptomatic COVID-19 patients demonstrated an elevated frequency of pDC IFN-α⁺ levels in absence of stimulation that vanished over time while, relatively, their ability to respond to SARS-CoV-2-infected cells increased at late time-points (Fig. 2h, left panel). A limited induction of pDC IFN-α⁺ in response to SARS-CoV-2-infected cells was observed in severe COVID-19 patients at all analyzed time points (Fig. 2h, right panel). We next performed a kinetic analysis for a severe COVID-19 patient, who had a high level of anti-IFN-α antibody detected in the blood (patient description included in Supplementary Table 1)[21–27]. This analysis showed that the ability of pDC to respond to stimulation was blunted in this patient (Fig. 2h, right panel; represented by stars with yellow-center). The kinetics of pDC IFN-λ⁺ response presented a similar pattern for both mild/asymptomatic and severe groups with basal level in absence of stimulation at early time-points and low response to SARS-CoV-2 infected cells, while pDC response recovered at a late time (Supplementary Fig. 2h), and this was paralleled by the IL-6⁺ frequency in HLA-DR⁺CD14⁺ population (Supplementary Fig. 2i).

Together our results demonstrated that the monocytic subsets likely contribute to an exacerbated pro-inflammatory response implying notably IL-6 production. The monocytic subsets do not produce IL-6 in response to incubation with infected cells in our ex-vivo coculture (Supplementary Fig. 2d), suggesting that IL-6 production by these cells might be via indirect activation and/or happening at a different time-point. As opposed, impaired IFN-I/λ response following cell contact between SARS-CoV-2-infected cells and pDCs from severe COVID-19 patients, including those with anti-IFN-α antibodies, suggested a 'silencing/unresponsive state' of pDCs in this context. This might be due to the lack of an amplification loop by ISG resulting in lower activated state and IFN-α production by pDCs and to some extent, similarly for IFN-λ1 expression by mDC1.

### Limited pDC differentiation and cytotoxic activity when in contact with SARS-CoV-2-infected cells

As we found that pDC activation is a salient feature that negatively correlates with COVID-19 severity, we aimed to determine how contact with SARS-CoV-2-infected cells impacts the varied downstream signaling and function of pDCs. First, we analyzed the expression of pDC surface molecules enabling the stimulation of adaptive responses, namely HLA-DR, an MHC class II cell surface receptor driving the activation of CD4⁺ T cells, and the B cell ligand CD70 expressed by pDCs and known to interact with CD27 and to induce proliferation and differentiation of B cells into plasmablast[59]. Our results showed that pDCs upregulated both surface molecules in response to contact with SARS-CoV-2-infected cells, although at a lower level compared to stimulation by a cell-free virus (influenza virus; flu) and synthetic agonists (Fig. 3a, b and Supplementary Fig. 3a).

Similarly, surface expression of CD83, an activation marker for antigen-presenting cells[60,61] was induced jointly with PD-L1 upon contact with SARS-CoV-2-infected cells (Fig. 3c and Supplementary Fig. 3a). SN from SARS-CoV-2-infected cells induced HLA-DR, CD83,

and PD-L1 expression, but not as potently as other soluble agonists. These results obtained with purified pDCs are in agreement with results obtained with pDCs present in PBMCs cocultured with SARS-CoV-2-infected cells (Fig. 1).

pDCs are now recognized to be a heterogeneous population composed of subsets endowed with diversified functions[28,34,35,62–65]. Importantly, stimulation of pDCs can impact the frequency and phenotype of these diversified subsets. We thus assessed the expression of a set of surface molecules previously assigned to define specific subpopulations of pDCs, i.e., CD2, CD5, AXL, CD80, and PD-L1[28,34,35,62–65]. CD2^hi pDCs have a survival advantage and are able to efficiently trigger the proliferation of naive allogenic T cells[63,64,66]. Stimulation of pDCs by contact with SARS-CoV-2-infected cells did not impact the frequency of CD2^hi pDCs, and this CD2^hi subset displayed an activation profile similar to the one of CD2^low subset (Supplementary Fig. 3b, c). CD2^hiCD5⁺AXL⁺ pDCs were defined as a subset that display limited IFN-I production capacity but can potently activate T cells, but represent a very scarce subpopulation of pDCs[65]. The CD2^hiCD5⁺AXL⁺ pDCs modestly decreased upon stimulation by SARS-CoV-2-infected cells, their SN or TLR agonists, yet whether this intriguing decrease is relative to the activated state of pDCs requires further investigation (Supplementary Fig. 3d). We then addressed the diversification of pDCs into functionally distinct populations defined by PD-L1/CD80 expression. In agreement with previous reports, the stimulation by cell-free agonists and influenza virus triggered the differentiation into all subsets: PD-L1⁺CD80⁻, PD-L1⁺CD80⁺, and PD-L1⁻CD80⁺ pDCs[34,35,62]. In sharp contrast, direct contact with SARS-CoV-2-infected cells restricted the differentiation of pDCs only towards the PD-L1⁺CD80⁻ subset (Fig. 3d).

Collectively, these results indicated that, unlike activation by cell-free viruses and agonists, direct activation of pDCs by SARS-CoV-2-infected cells restricted their differentiation into specific functional subsets, i.e., inducing a maturation only into PD-L1⁺CD80⁻ subset.

We observed that pDCs expressed PD-L1 (i.e., programmed cell-death ligand-1) when cocultured with SARS-CoV-2-infected cells. Therefore, we further examined the induction of the cytotoxic activity of pDCs. In keeping with the previously reported induction of membrane-bound TNF-related apoptosis-inducing ligand (mTRAIL) expression by pDCs upon viral stimulation (e.g., HIV)[67–69], we found that mTRAIL was readily co-expressed along with PD-L1 and CD83 upon stimulation by influenza virus and synthetic agonists (Fig. 3e). Nevertheless, mTRAIL upregulation was more limited upon pDC coculture with SARS-CoV-2-infected cells as compared to stimulation by cell-free stimulation (Fig. 3e, total % of the bars). Moreover, the frequencies of Annexin V⁺/7-ADD⁺ apoptotic Calu-3 and A549-ACE2 cells in cocultures with pDCs were similar between infected (i.e., pDC activation) and uninfected (i.e., no pDC activation) conditions, hence suggesting that mTRAIL expression on activated pDCs did not endow these cells with cytotoxic activity (Fig. 3f). In line with this, the contact with activated pDCs did not markedly impact the viability of the cocultured SARS-CoV-2-infected cells (i.e., when comparing condition with or without the anti-α_L integrin, which inhibits contact and pDC activation), nor the viability of activated pDCs themselves, even when analyzed after 48 h of coculture (Fig. 3g).

Overall, these results showed that activation of pDCs by SARS-CoV-2-infected cells did not induce cytotoxic activity and led to their preferential diversification into functional pDC subsets known to be specifically able to robustly produce IFN-I/λ.

### pDC activation by SARS-CoV-2-infected cells primarily leads to the antiviral state via IFN-I/λ production

We then sought to examine deeper the signaling pathways at play in pDCs upon activation by coculture with SARS-CoV-2-infected cells. TLR7-dependent activation of pDCs can induce a 'bifurcated' signaling leading to (1) IFN-I/λ production mostly via IRF7-related signaling and other cytokines and (2) activation/differentiation markers and

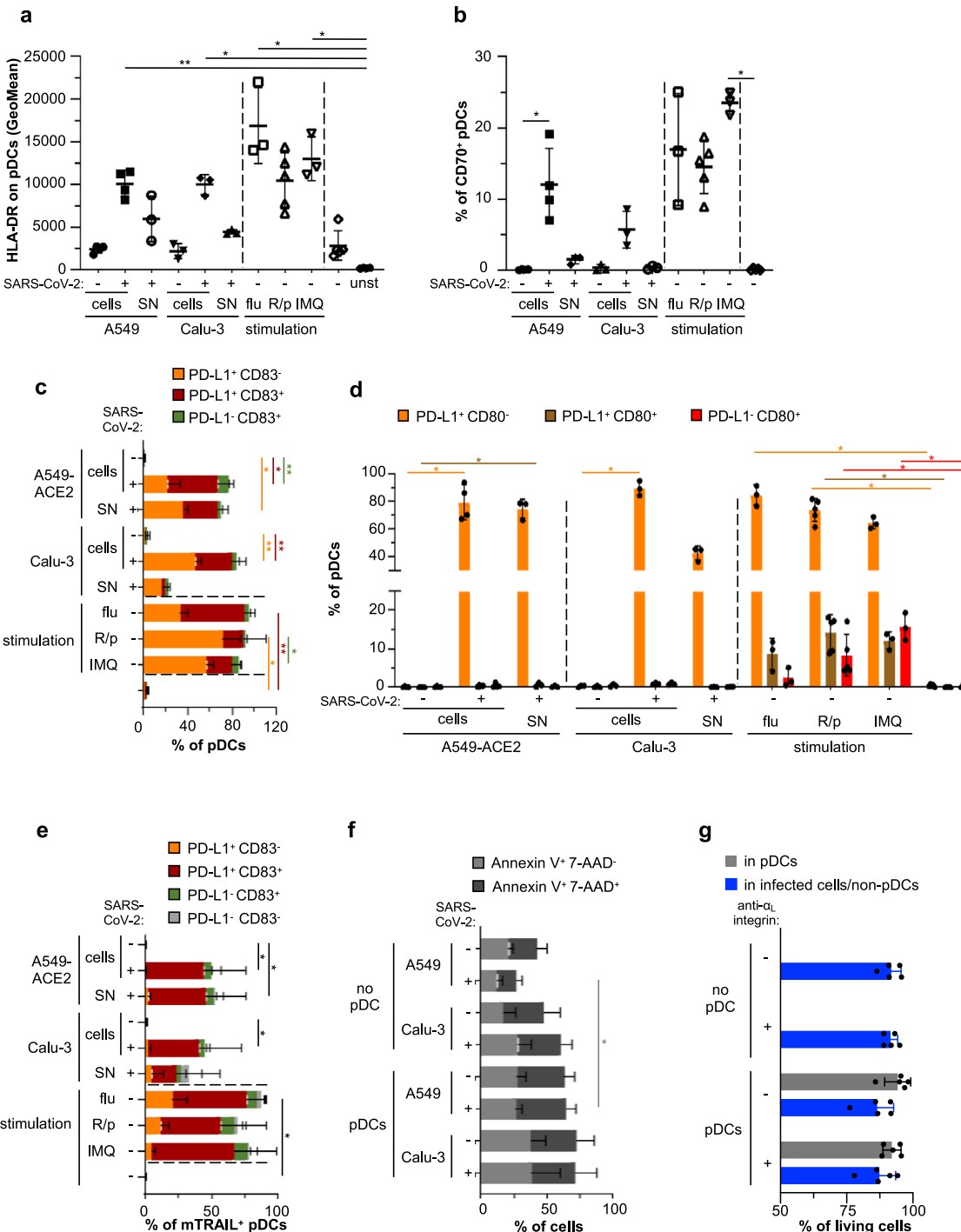

**Fig. 3 | SARS-CoV-2-infected cells induce pDC maturation and phenotypic diversification.** Human pDCs isolated from healthy donors were cocultured with SARS-CoV-2-infected [+] or uninfected [−] A549-ACE2 or Calu-3 cells (indicated as [cells]), or were stimulated with 100 μl of cell-free supernatants [SN] collected immediately prior coculture from the corresponding SARS-CoV-2-infected cells (viral titers ≈2.5 × 10⁵ ffu/ml · MOI ≈ 1/pDCs), or were stimulated with influenza virus [flu] (viral titers ≈ 10⁷ pfu/ml · MOI ≈ 0.5/pDCs), R848/polyI:C [R/p] or imiquimod [IMQ] for 14–16 h, [unst]; unstained. **a**–**e**, Quantification by flow cytometry of HLA-DR Geomean (**a**) or the frequency of pDCs positive for CD70 (**b**), PD-L1 and/or CD83 (**c**), PD-L1 and/or CD80 (**d**), PD-L1 and/or CD83 among mTRAIL⁺ pDCs (**e**) determined in gated pDCs (live cells⁺ singulets⁺ CD123⁺ BDCA-2⁺, see the gating strategy in Supplementary Fig. 3a). **f** SARS-CoV-2-infected [+] or uninfected [-] A549-ACE2 or Calu-3 cells were cultured alone [no pDC] or cocultured with pDCs for 14–16 h.

Quantification by flow cytometry of the frequency of gated A549-ACE2 and Calu-3 cells positive for Annexin V and/or 7-AAD. **g** icSARS-CoV-2-mNG-infected A549-ACE2 were cultured alone [no pDC] or cocultured with pDCs in the presence or absence of anti-αLintegrin blocking antibody for 48 h. Quantification by flow cytometry of the frequency of living cells using live-dead marker in the gated pDCs (stained with CellTrace Violet prior to coculture) and infected cells/non-pDC. Bars represent means ± SD; Each dot in **a**, **b**, **d** and **g**, represents one independent experiment using distinct healthy donors; **a**, **b**, **d**, *n* = 4 independent for pDC cocultured with A549-ACE2 cells and *n* = 3 with Calu-3 cells as; **c**, *n* = 3 for pDCs cocultured with A549-ACE2 cells and *n* = 4 with Calu-3 cells; **e** *n* = 5; **f**, *n* = 4; **g** *n* = 5. The data were analyzed using Kruskal–Wallis Global test and *p* values were calculated with Tukey and Kramer test; *≤0.05 and **≤0.005. The exact *p* values are indicated in Supplementary Fig. 8. Source data are provided as a Source Data file.

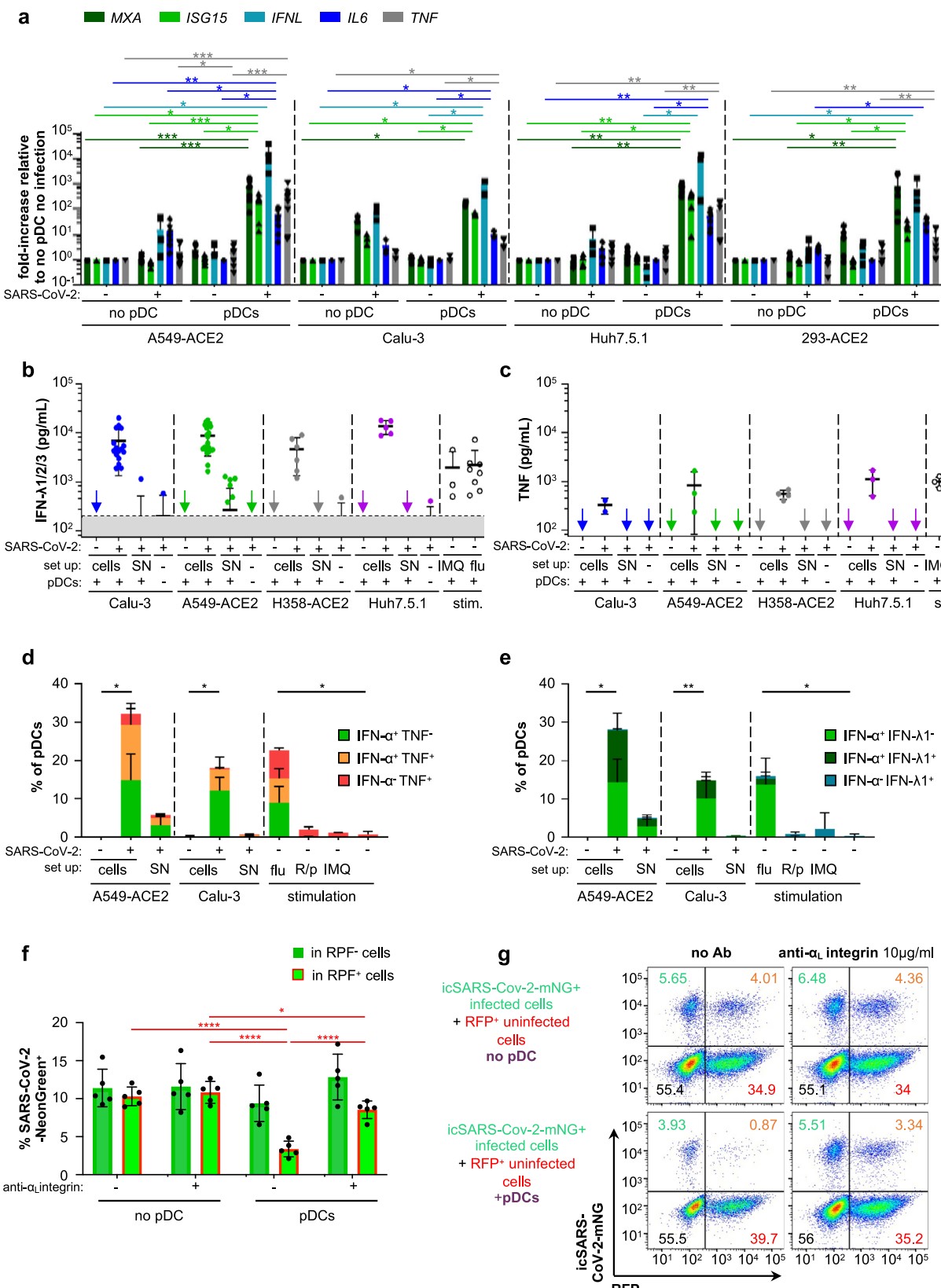

inflammatory cytokines via NF-κB-pathway[28,70]. We noticed that expression of the activation markers HLA-DR, CD70, CD83, mTRAIL by pDCs was weaker when induced by contact with SARS-CoV-2-infected cells compared to soluble agonist stimulation (Fig. 3). These markers/proteins are primarily regulated by NF-κB-mediated signaling (Supplementary Table 3)[45,46]. To further define the signaling active in pDCs,

we quantified transcripts levels of representative key effectors regulated by IRF7/IFN-I signaling (i.e., *MXA, ISG15,* and IFNλ) versus those primarily regulated by NF-κB-pathway (i.e., IL-6 and *TNF*) (Supplementary Table 3). Cocultures of pDCs and SARS-CoV-2-infected cells induced more the IRF7/IFN-I-regulated molecules than the representatives of the NF-κB-pathway (Fig. 4a). This was again confirmed by

**Fig. 4 | Cell-cell contact-dependent sensing of SARS-CoV-2-infected cells by pDCs induces a robust production of IFN-I/λ and other cytokines leading to the inhibition of viral spread.** Human pDCs isolated from healthy donors were cocultured with SARS-CoV-2-infected [+] or uninfected [−] cells or were incubated with 100 μl of cell-free supernatant [SN] collected immediately prior to coculture from the corresponding SARS-CoV-2-infected cells, or were stimulated by influenza virus or agonists (as in Figs. 1, 4). For all panels, when appropriated, the *p* values are indicated as follows: ≤0.05 as \*; ≤0.005 as \*\*; ≤0.0005 as \*\*\*; and ≤0.00005 as \*\*\*\*. The exact *p* values are indicated in Supplementary Fig. 8. **a** Quantification of the induction of ISG (*MXA*, *ISG15*), type III IFN (*IFNλ1*) and pro-inflammatory cytokines (*IL-6*, *TNF*) mRNAs in SARS-CoV-2-infected or uninfected A549-ACE2, Calu-3, Huh7.5.1 or 293-ACE2 cultured with [pDC] or without pDC [no pDC] by RT-qPCR; means ± SD; *n* = 8 independent experiments using distinct healthy donors in experiments using A549-ACE2 cells; *n* = 3 for Calu-3 cells; *n* = 6 for Huh7.5.1 cells; *n* = 5 for 293-ACE2; statistical analysis using Kruskal–Wallis Global test; *p* values as:

\* ≤0.05, \*\* ≤0.005 and \*\*\* ≤0.0005 (Tukey and Kramer test). **b, c** Quantification of IFN-λ1/2/3 (**b**) and TNF (**c**) in SN of pDCs cocultured with SARS-CoV-2-infected cells [cells] versus [SN], as indicated. Bars represent means ± SD; *n* = 3–14 (left panel) *n* = 2–4 (right panel) independent experiments using distinct healthy donors. **d, e** Quantification by flow cytometry of the frequency of pDCs positive for IFN-α and/or TNF (**d**) and IFN-α and/or IFN-λ1 (**e**). Bars represent means ± SD; *n* = 3–5 independent experiments using distinct healthy donors. **f, g** A549-ACE2 cells were infected by icSARS-CoV-2-mNG for 24 h and then cocultured with isolated pDCs for 48 h. Cocultured cells were treated or not with anti-$\alpha_L$ integrin blocking antibody (10 μg/mL). Viral transmission from icSARS-CoV-2-mNG$^+$-infected cells to RFP$^+$ uninfected cells in cocultures with pDCs or without pDCs [no pDC] was quantified by flow cytometry (**f**) and representative dot plots (**g**). Results are expressed as the percentage of cells positive for mNeonGreen (mNG$^+$) in the RFP$^+$ (orange numbers) and RFP$^−$ (green numbers) cell populations. Means ± SD; *n* = 4–5 independent experiments. Source data are provided as a Source Data file.

the quantification of secreted cytokines in the cocultures, demonstrating higher levels of IRF7/IFN-I-regulated IFN-λ1/2/3 compared to TNF (Fig. 4b compared to 4c). This observation was also in agreement with the high level of secreted IFN-α in similar experiments (Fig. 1a, b). Similar observations were made at different time points post-coculture (Supplementary Fig. 4b). Of note, no detectable cytokine and low level of the corresponding transcript expression were detected in SARS-CoV-2-infected cells in the absence of pDCs, or when pDCs were stimulated by SARS-CoV-2 SNs. As in these assays, the detection included expression by both pDCs and cocultured infected cells (Fig. 4a), the low activation level of the NF-κB-pathway and/or its variability among different infected cell types can be explained by a contribution of the infected cells, as a feedback loop of the response to activated pDCs. Hence, we assessed cytokine expression at single-cell level by flow cytometry in the gated pDCs (Fig. 4d, e and Supplementary Fig. 4a, c). Stimulation by SARS-CoV-2-infected cells elicited a higher frequency of IFN-α$^+$ pDCs compared to TNF$^+$ pDCs (Fig. 4d). Remarkably, almost all TNF$^+$ pDCs were also IFN-α$^+$ when stimulated by contact with SARS-CoV-2-infected cells, as opposed to the detection of TNF$^+$IFN-α$^−$ pDCs when stimulated by cell-free influenza virus (Fig. 4d). This was similarly observed at different time points post-coculture (Supplementary Fig. 4c). While the synthetic agonists induced IFN-α secretion by pDCs detected by ELISA as early as 4 h post-stimulation, yet this response is greatly lower at later time points as compared to pDC IFN-α production triggered by SARS-CoV-2-infected cells (Supplementary Fig. 4b, right panels). This might explain why in our experimental setting (i.e., unexpectedly and distinct from some other studies, e.g.,[33,34]), the pDC IFN-α production induced by synthetic agonists was below the detection limit by flow cytometry, as opposed to the robust response to SARS-CoV-2-infected cells or cell-free influenza virus (Supplementary Fig. 4c, left panels). In sharp contrast, and as validation of our experimental setting, synthetic agonists triggered a potent pDC TNF production, markedly higher compared to pDCs cocultured with infected cells (Supplementary Fig. 4c). Similar analysis performed for IFN-α$^+$ combined with IFN-λ1$^+$ also showed that virtually all IFN-λ1$^+$ pDCs were IFN-α$^+$ (Fig. 4e and Supplementary Fig. 4a). Again, supernatants from SARS-CoV-2-infected cells as well as the physical cell separation, yet allowing liquid diffusion prevented cytokine production by pDC at any time post-coculture (Fig. 3b–e and Supplementary Fig. 4b, c).

Collectively, our data provided evidence that, in contrast to stimulation with cell-free viruses and agonists, the pDC response to contact with SARS-CoV-2-infected cells is biased towards IRF7-mediated signaling that leads to a robust IFN-I/λ production.

## The pDC response controls SARS-CoV-2 spread and replication
As the response to SARS-CoV-2-infected cells primarily induced IFN-I/λ antiviral signaling, we next aimed to define how the pDC response

inhibits viral propagation. pDCs were cocultured for 48 h with icSARS-CoV-2-mNG-infected A549-ACE2 cells (mNG$^+$) and uninfected A549-ACE2 RFP$^+$ cells, and viral spread was quantified by flow cytometry. The results demonstrated that the pDC response readily prevented viral spread from mNG$^+$ infected cells to initially uninfected RFP$^+$ cells (Fig. 4f, g). The inhibition of contact between pDCs and infected cells via blockade of $\alpha_L$ integrin restored viral propagation to levels comparable to those measured in the absence of pDC (Fig. 4f, g). This indicated that the establishment of cell-contact via adhesion molecules is required for pDC-mediated antiviral response.

Interestingly, the impact of pDC antiviral response on cells infected prior to coculture (i.e., RFP$^−$mNG$^+$ infected cells) was lower as compared to the spread to uninfected cells (i.e., RFP$^+$) (Fig. 4f, g). This is likely owing to the inhibition of IFN-I/λ signaling by SARS-CoV-2 within infected cells[71,72]. Therefore, we hypothesized that the reduction could be more potent if infected cells were directly in contact with pDCs as such contact could allow a concentrated antiviral response toward the infected cell. To test this hypothesis, we established an assay of 24 h-long live-imaging of the coculture of pDCs (stained with CM-Dil; red) and icSARS-CoV-2-mNG infected cells (mNG$^+$; green) using spinning-disk confocal microscopy. As depicted by the example of time-sequence imaging (Fig. 5a and Supplementary Fig. 5a), pDC contact with mNG$^+$ infected cells lead to a control of viral replication in the targeted infected cells, reflected by the decreased mNG fluorescent reporter signal. We controlled that these infected cells, although not mNG$^+$ anymore, were still physically present by using enhanced fluorescent signal analysis (Fig. 5a and Supplementary Fig. 5a; lower panels). Viral control by pDCs was seen for pDC/infected cell contacts starting at different time points in the course of the coculture (i.e., up to 14 h after record onset, Supplementary Fig. 5d). Similar analysis was performed for several infected cells, either in contact with a pDC versus not in contact, showed that the decrease in viral replication (i.e., mNG fluorescence intensity) was observed only for infected cells directly in contact with pDCs (Fig. 5b–d and Supplementary Fig. 5b). Of note, live-tracking of mNG fluorescence intensity performed in infected cells cultured without pDCs, provided the basal level of the mNG fluorescence intensity (done simultaneously with record of coculture with infected cells). This basal level was comparable to the one measured in infected cells cocultured with pDCs that were not in direct contact with infected cells (Fig. 5c, d and Supplementary Fig. 5c). The decrease in mNG fluorescence intensity were detected several hours after the onset of contact, likely owing to the time-window needed to inhibit viral replication (Fig. 5e and Supplementary Fig. 5e).

To assess that the mNG reporter reflects the replication level, we combined it with the detection of other viral replication parameters, i.e., the dsRNA reflecting the replication intermediate species and the Spike protein analyzed by both Flow cytometry and Confocal imaging analysis. The results demonstrated that mNG$^+$ cells also express dsRNA

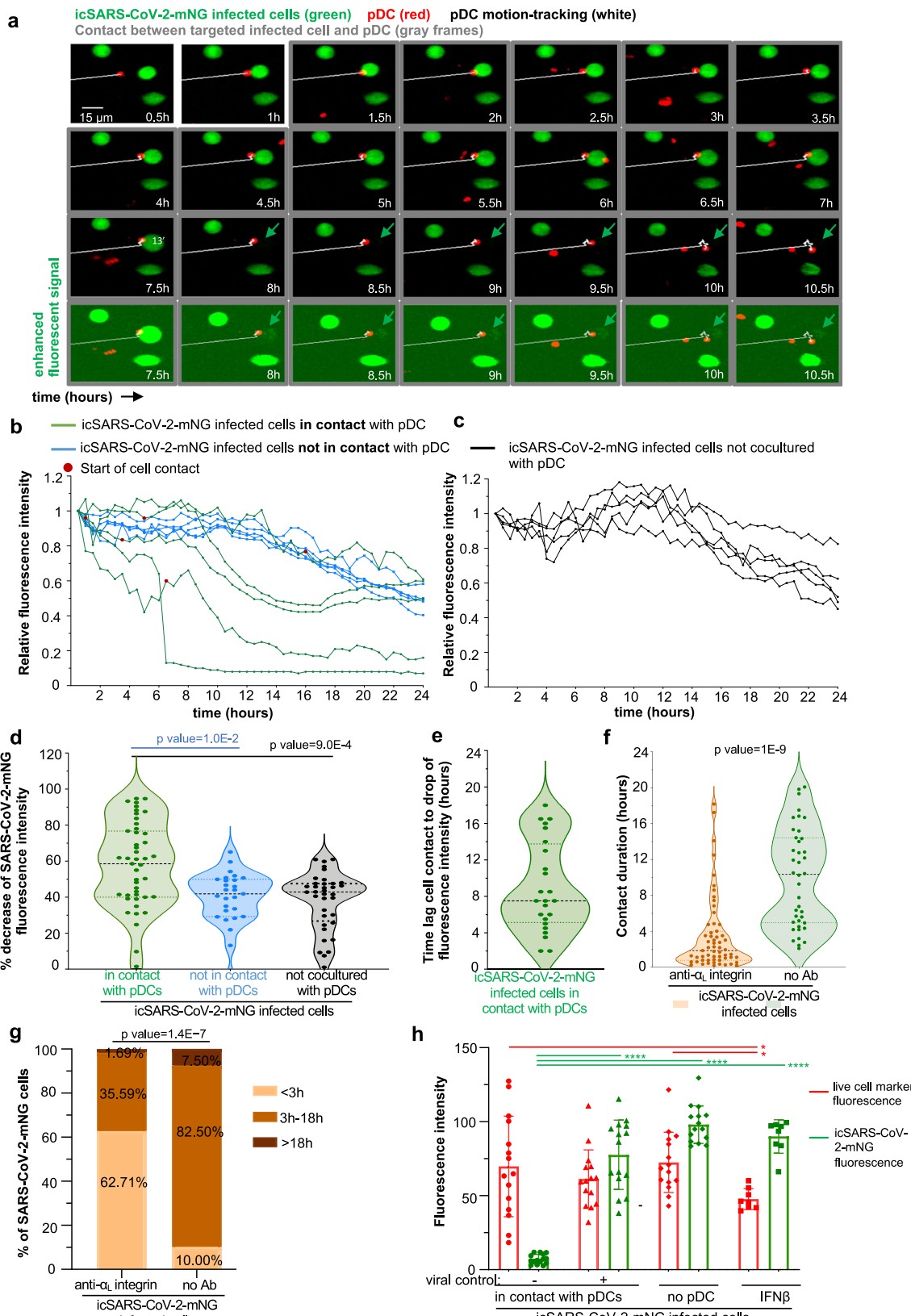

and/or Spike protein (Supplementary Fig. 6). This was observed for the majority mNG⁺ cells by confocal analysis (Supplementary Fig. 6a–c) and even detected for virtually all mNG⁺ cells when assessed by confocal imaging analysis (Supplementary Fig. 6d–g). As opposed, mNG⁻ cells were also dsRNA⁻ and/or Spike⁻. These observations were further confirmed when focusing the quantification to icSARS-CoV-2-mNG-

infected cells nearby pDCs [contact] as defined for pDC/infected cell distance inferior to 5 µm (Supplementary Fig. 6d–f). These results demonstrated that mNG reporter reflects the replication level.

Our results further showed that inhibition by anti-αL integrin greatly reduces the duration of contact between pDCs and infected cells, as compared to untreated coculture (Fig. 5f, g). The inhibition of

**Fig. 5 | Targeted antiviral activity of pDCs toward SARS-CoV-2-infected cells.** Live imaging of coculture of icSARS-CoV-2-mNG-infected cells with pDCs by spinning-disk confocal analysis. A549-ACE2 cells were infected by icSARS-CoV-2-mNG for 48 h prior to coculture with pDCs. **a** Representative time sequence of pDCs (CM-DiI stained, red), tracked using motion automatic tracking plug-in in image J (white line) in contact with icSARS-CoV-2-mNG-infected cells (green arrow). The time points when pDCs are in contact with infected cells are framed in gray. Bottom panels show same imaging of the seven last time points with enhanced fluorescence signal. **b**–**d** Calculation of mNeongreen fluorescence intensity over time in individual icSARS-CoV-2-mNG-infected cells cocultured with pDCs and in contact (green) *versus* cocultured with pDC but not in contact (blue) and, as control/reference, in simultaneously recorded cultures of icSARS-CoV-2-mNG-infected cells in absence of pDC [no pDC, black]. The mNeongreen fluorescence intensity is determined using area-integrated intensity and mean value (quantification tools in Image J). **b**, **c** Representative kinetic analysis of mNeongreen fluorescence intensity in individual icSARS-CoV-2-mNG-infected cells cocultured with pDCs (**b**) in contact or not as indicated *versus* in absence of pDC (**c**). The time point corresponding to the onset/start of contact is indicated by a red dot. The results are presented as the mNeongreen fluorescence intensity at the indicated time relative to time 0 of record set to 1; *n* = 5 individually recorded cells analyzed per condition from one representative experiment (and *n* = 10–12 in other experiments). **d** Violin plot representation of the decrease of mNeongreen fluorescence intensity (percentage). Each dot represents one infected cell (*n* = 118); 4 independent experiments. Statistical analysis was performed using ANOVA (test global) and Tukey multiple comparisons of means. **e** Violin plot representations of time-lag between the onset of pDC contact and the decrease of mNeongreen fluorescence intensity defined as

>50% of the initial fluorescence intensity. Each dot represents one individual icSARS-CoV-2-mNG-infected cells in contact with pDCs, *n* = 25 from four independent experiments. **f**, **g** Live imaging of coculture of icSARS-CoV-2-mNG-infected cells with pDCs treated with blocking antibodies against αL-integrin (10 μg/mL, added 15 min prior and kept during the coculture) versus not treated coculture. The contact duration between icSARS-CoV-2-mNG-infected cells and pDCs was determined for individual contact and presented in violin plot (**f**) and as categories of contact assigned as short-duration (<3 h) versus long-duration (3–18 h and >18 h) (**g**). Statistical analyses were performed using Wilcoxon test (**f**) and Fisher's test (**g**). **h** The icSARS-CoV-2-mNG-infected cells were stained with a fluorescent live cell marker prior to coculture with pDCs and live imaging by spinning-disk confocal analysis. Calculation of the fluorescence intensity of both the live cell marker and mNeongreen over time in individual icSARS-CoV-2-mNG-infected cells cocultured and in contact with pDCs, leading to control of viral replication [+], defined as decrease of fluorescence intensity > 50% relative to the initial mNG fluorescent intensity or not [−]. The simultaneous record of cultures of icSARS-CoV-2-mNG-infected cells in the absence of pDC [no pDC] and treated with recombinant IFNβ (100 UI/mL) served as control/reference. The results are presented as the fluorescence intensity of mNeongreen (green bars) and living cell marker (red bars) relative to the levels in individual icSARS-CoV-2-mNG-infected cells prior to pDC contact and at 30 min-record and set to 100. Each dot represents one individually recorded cells, *n* = 15 analyzed per condition and means ± SD; *n* = 2 independent experiments. Statistical analyses were performed using pairwise comparisons using Wilcoxon rank-sum exact test and *p* values adjustment method: fdr and *p* values are indicated: ≤0.05 as *; and ≤0.00005 as ****. Source data are provided as a Source Data file.

cell adhesion molecule results in a majority of short-duration contact (*i.e.*, shorter than 3 h, Fig. 5g). Therefore, in accordance with decrease of viral replication occurring upon sustained duration of contacts (Fig. 5e and Supplementary Fig. 5e), the anti-αL integrin restored an efficient viral spread even in presence of the pDCs, as demonstrated by flow cytometry analysis (Fig. 4f, g).

We further performed side-by-side analyses of the levels of cell viability (i.e., live cell marker) and viral control (i.e., mNG fluorescent reporter) at single-cell level and in association with the tracking of pDC contact with infected cells (Supplementary Fig. 5f). The results demonstrated that the control of viral replication (i.e., mNG fluorescent reporter, green bars in Fig. 5h and Supplementary Fig. 5g) is not strictly explained by cell death of the targeted infected cells, as the intensity of the live cell marker was comparable when viral replication was inhibited or not (i.e., live cell marker, red bars in Fig. 5h and green curves in Supplementary Fig. 5h). The levels of live cell markers were comparable in infected cells cocultured or not with pDCs (Fig. 5h and Supplementary Fig. 5h, i), while the intensity of this live cell marker diminished upon addition of recombinant IFN-β (Fig. 5h and Supplementary Fig. 5j). These results are in accordance with flow cytometry analysis of the cell-death marker expressions and living cells (Fig. 3e–g).

Overall, these results demonstrated that pDCs established sustained contact with SARS-CoV-2 infected cells via $\alpha_L\beta_2$ integrin/ICAM-1 adhesion complex leading to an efficient antiviral response directed toward the infected cells that shut down viral replication.

## Discussion

Here we demonstrate that pDCs are the key mediators of the IFN-I/λ response against SARS-CoV-2-infected cells. Importantly, our study of immune cells from COVID-19 patients at the single-cell and functional levels establishes that the pDC response is pivotal to control COVID-19 severity. Especially, the sensing of SARS-CoV-2-infected cells is defective in patients with severe disease. As opposed, in healthy donors the scanning function of pDCs for immune surveillance operates via the establishment of sustained contacts with SARS-CoV-2-infected cells *by* cell adhesion molecules. This sensing induces IRF7/IFN-I/λ-prioritized signaling in pDCs, while leaving inactive the NF-κB-mediated pathway. Next, the pDC-mediated IFN-I/λ response is specifically targeted

towards SARS-CoV-2 infected cells. This specialized function thus enables pDCs to efficiently turn-off viral replication, likely owing to a concentrated efflux of antiviral effectors at the contact site with infected cells (Fig. 6).

Insights on several hallmarks of IFN-I/λ pathway as being critical in COVID-19 severity are emanating from recent publications. First, neutralizing autoantibodies against several cytokines and genetic defects affecting the IFN-I pathway have been identified in life-threatening COVID-19, as autosomal disorders of IFN-I immunity and autoantibodies underlie at least 10% of critical COVID-19 pneumonia cases[21–27,73]. Second, patients with severe diseases exhibit reduced circulating pDCs and low plasma IFN-I/λ levels when compared to mild/asymptomatic COVID-19 patients[7,19,20], yet the impact of pDC response on the severity is still elusive in other publication[74]. Third, IL-3, which increases innate immunity likely by promoting the recruitment of circulating pDCs into the airways, is reduced in the plasma of patients with high viral load and severity/mortality[75]. This evidence highlighted that several biomarkers of IFN-I/λ response are diminished in COVID-19 patients, and hereby pointed to pDCs as a primary candidate in the progression of COVID-19. Here, using an original approach to study innate immunity in ex-vivo-stimulated PBMCs from COVID-19 patients, we report that the functionality of pDCs is markedly blunted in severe patients, as observed upon their stimulation by contact with SARS-CoV-2-infected cells as well as other TLR agonists. Nonetheless, owing to the technical challenge to perform these functional analyses for a larger cohort of patients, future investigations including a larger diversity of groups (e.g., additional patients with anti-IFN antibody, with immunosuppressive treatments, children etc.) will enable to reach definitive conclusion across diverse human populations.

pDCs comprise distinct subpopulations capable of varied functions and efficacy levels to mount an IFN-I/λ response[28,34,35,62–65,76,77]. Viral infections and attendant inflammation potentially impact the frequency and functionality of the distinct pDC subpopulation including effect on pDC renewal[62,76,78]. Such modulations could be imprinted either by the micro-environment via crosstalk with immune cells, or by pDC activation itself.

We now report that neither pDC subpopulation frequencies nor their distinct ability to respond to SARS-CoV-2 is impacted by stimulation and contact with SARS-CoV-2-infected cells. This is notably

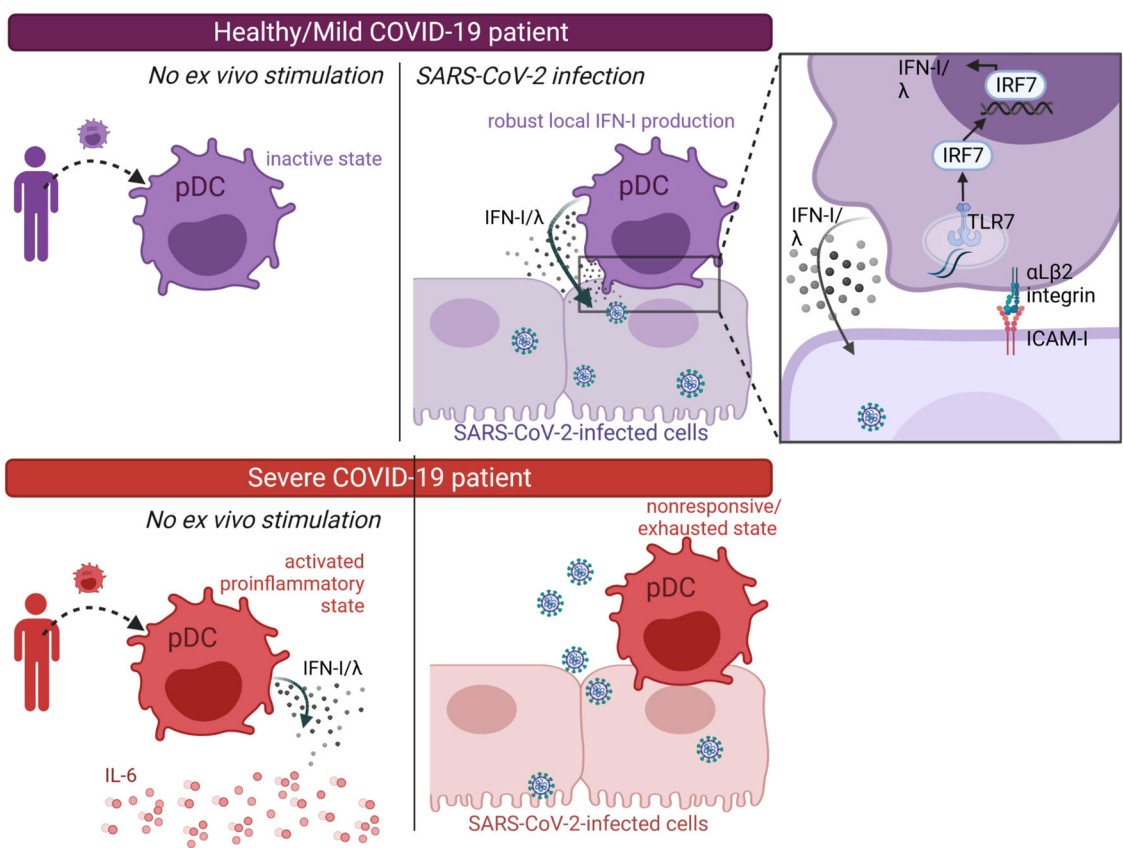

**Fig. 6 | Proposed model of the innate responses associated with COVID-19 severity.** Our longitudinal study of the innate responses by ex-vivo stimulation of PBMCs from COVID-19 patients, and across distinct disease severities (i.e., mild/asymptomatic versus severe COVID-19 and healthy donors, as reference) highlighted the following proposed model. pDCs from mild/asymptomatic patients and healthy donors (in purple) robustly produce IFN-I/λ upon cell contact with SARS-CoV-2-infected cells (upper panel). As opposed, pDCs from severe patients (in red) produce IFN-I/λ in absence of ex vivo stimulation, but fail to be activated by contact with SARS-CoV-2-infected cells (lower panel). This non-responsive/exhausted state

of pDCs in severe COVID-19 patients is associated with an elevated level of pro-inflammatory cytokines (here represented by IL-6, red round symbols) that are most likely produced by the HLA-DR⁺CD14⁺ monocytes. As shown on the zoomed view of the contact site (right panel, at the top), the short-range sensing of SARS-CoV-2-infected cells by pDCs requires cell contact mediated by adhesion complexes, identified as $\alpha_L\beta_2$ integrin and ICAM-1. This triggers TLR7-induced signaling via IRF7 leading to an IFN-I/λ-prioritized response while leaving inactive the NF-κB-mediated signaling.

---

illustrated for CD2^low/CD2^hi pDCs that display a similar activation profile upon stimulation with SARS-CoV-2-infected cells.

Interestingly, when comparing stimulations by contact with SARS-CoV-2-infected cells *versus* cell-free activators (i.e., viral particles and synthetic agonists), we found that the differentiation of pDCs defined by CD80 and/or PD-L1 expression is readily distinguishable depending on the TLR7 inducers. Whilst pDCs diversify towards all the different PD-L1/CD80 subsets upon cell-free stimulation (agonist and virus) in agreement with prior publications[34,35,62], the contact with infected cells restricts their evolution to PD-L1⁺CD80⁻ subsets. Of note, previous reports suggested that PD-L1⁺CD80⁻ pDCs are more efficient for IFN-I response compared to the other subsets[34,62,76].

In accordance with this phenotype, contact with SARS-CoV-2-infected cells induces an IRF7/IFN-I/λ-prioritized signaling in pDCs, while leaving inactive the NF-κB-mediated pathway. The 'bifurcated' signaling in TLR7-activated pDCs can occur independently toward either IFN-I/λ production or NF-κB activation leading to pro-inflammatory cytokines and activation markers. This bifurcation of signaling is expected to be dependent on the sub-cellular compartment in which the TLRs encounter activating signal: activation in early endosomes induces IFN-I while activation in endolysosomes triggers NF-κB-mediated signaling[28,79]. We previously reported that, in the context of other viral infections (i.e., various RNA genome viruses such as Dengue, Chikungunya, Hepatitis C, Zika viruses), contact with infected cells similarly induced a IRF7/IFN-I/λ prioritized signaling in

activated pDCs, as opposed to incubation of pDCs with cell-free viruses[49,70]. More recently, similar features of pDC activation upon cell contact were also elegantly observed for a DNA genome virus, the human cytomegalovirus[80]. Of note, we also show that stimulation of pDCs by cell-free SARS-CoV-2 supernatants induced the upregulation of some activation markers (e.g., HLA-DR, PD-L1), in accordance with other reports[34,35,62], but elicited virtually no IFN-I/λ response. We propose that the signaling downstream of TLR7, including the phosphorylation cascade might be impacted by cell polarity and physical contact with infected cells[81,82]. Further studies will aim at addressing this question.

Importantly, our results suggest that pDC response controls viral replication primarily in the infected cells in direct contact. This viral control does not seem to involve cell death of the targeted infected cells. We thus proposed that IFN-I/λ response is concentrated at the contact site and thus potent to induce host antiviral effectors along with other regulations of host pathways expected to occur in response to IFN-I/λ signaling, e.g., translation shutdown and modulation of transcriptional activity. All of this host changes can be at play to robustly control viral replication in the targeted infected cells.

Progression to severe COVID-19 predominates in elderly patients, since advanced age is a factor suspected to aggravate disease progression and to weaken innate immunity, potentially including pDC responsiveness[83,84]. This is in accordance with the demographic analysis of our COVID-19 cohort, as the age-ranges were 63.5-year-old

[interquartile of 48.0–73.0] and 40.5-year-old [24–57] for severe and mild/asymptomatic patients, respectively. The patient history and genetic factors can also explain the differential ability of pDCs from patients to mount a response against SARS-CoV-2-infected cells, as illustrated by the heterogeneous immune responses of patients, with critical anomalies in the functionality of pDC in severe COVID-19. Our kinetic analysis performed on a COVID-19 patient that presents auto-antibodies against IFN-I[21–23,25] shows that the responsiveness of their pDCs to stimulation by SARS-CoV-2-infected cells was blunted (Fig. 2h, Supplementary Fig. 2g, h). A future study is needed to expand this interesting preliminary observation, which already indicates that the IFN-I/λ response can reinforce via a positive feedforward regulation of the pDC antiviral function.

We demonstrated a comparable antiviral response by pDCs when performed using PBMCs versus isolated pDCs from healthy donors in different types of analyses and for various parameters in Fig. 1a–d and Fig. 2c–h (PBMCs) versus Figs. 3, 4 and Supplementary Figs. 3, 4 (isolated pDCs). The comparison using PBMCs from patients has the advantage to provide insights into the relative contributions of the different hematopoietic cell types in the same patient samples. The comparison of responses across patients with different COVID-19 severities suggested that pDC response to SARS-CoV-2-infected cells inversely correlates with an exacerbated inflammatory response (as illustrated by IL-6 production by monocytic cells) and a basal level of IFN-I/λ and inflammation activity. It is tempting to speculate that the progressive enrichment of pro-inflammatory cytokines in the lung micro-environment can imprint pDC responsiveness. In this scenario, an excessive elicitation of pDCs would lead to their functional 'silencing'. Albeit not formally demonstrated, the 'exhausted' pDCs in vivo might not be explained by prior exposure to SARS-CoV-2 infected cells since viral loads were similar in patients of mild/asymptomtic and severe groups, whilst the cytokine micro-environment was greatly distinct across disease severity. Future studies will be needed to elucidate the underlining mechanism that could lead to such 'exhausted' pDCs, as reminiscent of the findings of impaired pDC function in a distinct infection context[78]. In turn, the deficit of the antiviral control at the infected site owing to *exhausted* pDCs can then feedback as fuel for uncontrolled viral replication leading to more lung inflammation. Altogether our results thus highlight possible cross-regulation between immune cells in the course of COVID-19.

SARS-CoV-2 is a still-ongoing worldwide health threat, currently causing a significant human and economic burden, being exacerbated by a divergence into more severe variants. Here, we provide compelling evidence that pDCs are a key cell type in the initiation of antiviral responses against SARS-CoV-2. Further, this study identified the failure of pDC response as critical in COVID-19 severity. Moving forward our finding shall provide guidelines for predictive biomarkers, as associated with pDC responsiveness to SARS-CoV-2 infections and background IFN-I/λ response at different stages of disease. Furthermore, strategies to boost the pDC response, and especially their recruitment to the lung, can lead to the development of potential therapeutics against pulmonary viral infections.

## Methods
We confirm that our research complies with all relevant ethical regulations and validated by the national review boards for biomedical research (i.e., Comité de Protection des Personnes Sud Méditerranée I, Marseille, France) in April 2020 (ID RCB 2020-A00932-37); the French National Data Protection Agency under the number 20-097. This was also approved by an ethical committee for biomedical research (i.e., Comité de Protection des Personnes HCL) under the number 20-41 and in agreement with the General Data Protection Regulation (EU regulation 2016/679 and Directive 95/46/EC) and the French data protection law (Law n°78-17 on 06/01/1978 and Décret n°2019-536 on 29/05/2019). The '*Etablissement Français du sang*' (EFS) according to

standardized procedures for blood donation approved by the EFS Committee and followed provisions of articles R.1243–49 and the French public health code to obtain written non-opposition to the use of donated blood for research purposes from healthy volunteers.

### Preparation of viral stocks and infections
The clinical isolate was obtained from patients referenced in the GISAID EpiCoVTM database: as BetaCoV/France/IDF0571/2020; accession ID: EPI_ISL_411218[85], kindly provided by Dr. B. Lina. The infectious clone-derived mNeonGreen SARS-CoV-2 (referred to as icSARS-CoV-2-mNG) was kindly provided by Dr. Pei-Yong Shi and generated by introducing mNeonGreen into ORF7 of the viral genome[47]. Viral stock of Influenza A Virus (Flu A/H1N1/New caledonia; infectious titer of ≈10^7 plaque-forming unit (PFU)/ml)[86] was produced as previously described and kindly provided by Dr. V. Lotteau (CIRI, Lyon France).

### Cell lines and primary cell cultures
SARS-CoV-2-infected cells included the human alveolar basal epithelial cell lines, Calu-3 cells (ATCC HTB-55), A549 cells (ATCC CCL-185), and NCI-H358 cells (ATCC CRL-5807), Huh7.5.1 cells[87] (derived from Huh7 cells−kindly provided by Dr. F.V. Chisari; The Scripps research institute, San Diego), HEK-293 cells (ATCC CRL-1573) and Vero E6 cells (ATCC CRL-1586; kindly provided by Dr. M Bouloy; Institut Pasteur). The A549 cells, NCI-H358 cells, and HEK-293 cells were transduced to stably express the human angiotensin-converting enzyme 2 (ACE2; accession number: NM_021804) using a lentiviral vector, as previously described[87,88]. A549 cells were maintained in Roswell Park Memorial Institute (RPMI) 1640 Medium (Life Technologies) supplemented with 10% FBS, 100 units (U)/ml penicillin, 100 mg/ml streptomycin, and non-essential amino acids (Life Technologies) at 37 °C/5% $CO_2$. The NCI-H358 and Huh7.5.1 cells were maintained in Dulbecco's modified Eagle medium (DMEM) (Life Technologies) with the same supplements and 2 mM L-glutamine. Calu-3 cells, which endogenously express ACE2[89,90], along with TMPRSS2[91], were maintained in DMEM/Nutrient Mixture F-12 Ham (1:1) (Life Technologies) supplemented with gluta-MAX, 10% FBS, 100 units (U)/ml penicillin, and 100 mg/ml streptomycin. All cell were maintained at 37 °C/5% $CO_2$.

pDCs were isolated from 450 ml of blood units obtained from adult human healthy donors and according to procedures approved by the "Etablissement Français du sang" (EFS) Committee. PBMCs were isolated using Ficoll-Hypaque density centrifugation. pDCs were positively selected from PBMCs using BDCA-4-magnetic beads (MACS Miltenyi Biotec) and cultured as previously described[49]. PBMCs and pDCs were cultured in RPMI 1640 Medium (Life Technologies) supplemented with 10% FBS, 100 units (U)/ml penicillin, 100 mg/ml streptomycin, 2 mM L-glutamine, non-essential amino acids, 1 mM sodium pyruvate and 10 mM Hepes (Life Technologies) at 37 °C/5% $CO_2$.

### Cohort of SARS-CoV-2 infected patients
The constitution of the cohort i.e., symptomatic healthcare workers (COVID-SER patients) and patients admitted to ICU positive for COVID-19 (COVID-rea patients) was done by the collaboration of the Hospices Civils de Lyon (HCL), France. The participants were recruited at the Hospital (Hospices Civils de Lyon, Lyon, France) without criteria of exclusion other than pregnancy. Written informed consent was obtained from all participants and approval was obtained from the national review board for biomedical research in April 2020.

COVID-SER patients: For the mild adult COVID-19 cohort, the clinical study is registered on ClinicalTrial.gov (NCT04341142). In the present study, only patients with mild symptoms of COVID-19 were included. Written informed consent was obtained from all participants and approval was obtained from the national review board for

biomedical research in April 2020 (Comité de Protection des Personnes Sud Méditerranée I, Marseille, France; ID RCB 2020-A00932-37).

COVID-rea patients: blood samples were collected from COVID-19 patients hospitalized at the University Hospital of Lyon (Hospices Civils de Lyon), France. Diagnosis of COVID-19 was confirmed in all patients by RT-PCR. All critically ill patients, admitted to ICU, were included in the MIR-COVID study. This study was registered to the French National Data Protection Agency under the number 20-097 and was approved by an ethical committee for biomedical research (Comité de Protection des Personnes HCL) under the number 20-41. In agreement with the General Data Protection Regulation (Regulation (EU) 2016/679 and Directive 95/46/EC) and the French data protection law (Law n°78-17 on 06/01/1978 and Décret n°2019-536 on 29/05/2019), we obtained consent from each patient or his next of kin. These COVID-SER and COVI-rea cohorts thus consist of patient groups recognized by clinicians as: (i) patients admitted in intensive care units for severe disease at hospital admission (i.e., acute respiratory distress syndrome or severe pneumonia requiring mechanical ventilation, sepsis, and septic shock) are referred to as severe patients and (ii) patients with asymptomatic or mild symptoms (i.e., low-grade fever, cough, malaise, rhinorrhea, sore throat) are referred to as the group of mild/asymptomatic early when collected in the first 2 weeks and group of mild/asymptomatic late for later time points. A detailed description of the patient information along with the levels of the IFN-I/III signature and viremia, as determined in blood and nasal swab samples, respectively, is provided in Table 1 and Supplementary Table 1.

Blood samples from *healthy donors* were used as references and experimentally processed similarly. These samples were obtained from the national blood service, called '*Etablissement Français du sang*' (EFS) according to standardized procedures for blood donation approved by the EFS Committee and followed provisions of articles R.1243–49 and the French public health code to obtain written non-opposition to the use of donated blood for research purposes from healthy volunteers. The personal data were deidentified before transfer to our research laboratory. We obtained the favorable notice of the local ethics committee (Comité de Protection des Personnes Sud-Est II, Bâtiment Pinel, 59 Boulevard Pinel, 69 500 Bron) and acceptance from the French ministry of research (Ministère de l'Enseignement supérieur, de la Recherche et de l'Innovation, DC-2008-64) for the handling and conservation of these samples. To limit the risk of inclusion of asymptotic healthy donor: (i) part of the blood samples was collected prior to the pandemic and (ii) for blood collected during the SARS-CoV-2 pandemic, systemic examination and questioning/interview of the donors were performed and included symptoms, prior contacts at risk and vaccination, Thus, blood samples were excluded from our study if ongoing and/or recent COVID-positivity was suspected.

## Reagents
The antibodies used for immunostaining are listed in Supplementary Table 4. Ficoll-Hypaque (GE Healthcare Life Sciences). Other reagents included LPS, TLR3 agonist (Poly(I:C); LMW) and TLR7 agonist (R848 and Imiquimod) (Invivogen); TLR7 antagonist (IRS661, 5′-TGCTT GCAAGCTTGCAAGCA-3′ synthesized on a phosphorothionate backbone; MWG Biotech); mouse anti-$\alpha_L$ integrin (clone 38; Antibodies Online); mouse anti-ICAM-1 (Clone LB-2; BD Bioscience): Arp2/3 complex inhibitor I (CK-666; Merck Millipore); Fc Blocking solution (MACS Miltenyi Biotec); Golgi-Plug, cytoperm/cytofix and permeabilization-wash solutions (BD Bioscience); IFN-α and IFN-λ1/2/3 ELISA kit (PBL Interferon Source); IL-6 and TNF ELISA kit (Affymetrix, eBioscience); 96-well format transwell chambers (Corning); IL-6 and IFN-λ by U-PLEX Custom Human Cytokine assay (Meso Scale Diagnostics, Rockville, MD); 96-Well Optical-Bottom Plates (Thermo Fisher Scientific); cell-labeling solution using CellTrace Violet Cell Proliferation Kit (Life Technologies ref # C34557, C34571), Live/Dead Fixable Dead Cell Stain

Near-IR (Life Technologies ref #10119); Fixable Viability Dye eFluor 450 (Life Technologies); Zombie Aqua and Zombie Green Fixable Viability Kits (Biolegend); FITC Annexin V Apoptosis Detection Kit with 7-AAD (Biolegend); cDNA synthesis and qPCR kit (Life Technologies); poly-L-lysin (P6282, Sigma-Aldrich).

## Quantification of SARS-CoV-2 level in nasal swab samples of infected patients
Nucleic acid extraction was performed from 0.2 mL naso-pharyngeal swabs using NUCLISENS easyMAG and amplification was performed using Biorad CFX96. Quantitative viral load was determined using four internally developed quantification standards (QS) targeting the SARS-CoV-2 N gene: QS1 to QS4, respectively, at $2.5 \times 10^6$, $2.5 \times 10^5$, $2.5 \times 10^4$, $2.5 \times 10^3$ copies/mL of a SARS-CoV-2 DNA standard. These QS were controlled and quantified using the Nanodrop spectrophotometer (ThermoFisher) and Applied Biosystems QuantStudio 3D Digital PCR. In parallel, naso-pharyngeal swabs were tested using the CELL Control R-GENE kit (amplification of the HPRT1 housekeeping gene) that contains two quantification standards QS1 and QS2, at $10^4$ copies/µL (50,000 cells/PCR i.e. $1.25 \times 10^6$ cells/mL in our conditions) and $10^3$ copies/µL (5000 cells/PCR, i.e., $1.25 \times 10^5$ cells/mL in our conditions) of DNA standard, respectively, to normalize the viral load according to the sampling quality (Eq. 1).

$$Normalized\ viral\ Load[\log_{10} cp/mL]$$
$$= \log_{10}\left[\frac{number\ of\ SARS-CoV-2\ copies\ per\ mL}{number\ of\ cells\ per\ mL} \times 10^6 cells\ per\ mL\right] \quad (1)$$

## Analysis of IFN-I/λ signatures in blood samples of SARS-CoV-2-infected patients
**Transcript levels using Nanostring technology.** Targeted transcripts of the IFN-I pathway included SIGLEC1, IFI27, IFI44L, IFIT1, ISG15, RSAD2, HPRT1, POLR2A, and ACTB[92]. Transcript levels were determined using RNA extracted from the patient's blood samples by Maxwell 16 LEV simply RNA Blood kits that comprised an individual DNAse treatment[92]. Next, total RNA was eluted in 40 µL RNAse-free water, and concentration was quantified by spectrophotometry using a NanoVue (Biochrom). Only 200 ng of RNA were needed to achieve IFN signature with Nanostring technology. For the Elements system, capture (probe A) and tag (probe B) probe DNA oligos were designed by NanoString and synthesized by Integrated DNA Technologies (IDT). After hybridization at 67 °C for 18–20 h, samples were analyzed using the "High Sensitivity" protocol option on the nCounter Prep Station and counted on nCounter Digital Analyzer using maximal data resolution. Data were processed with nSolver version 4.0 software (NanoString Technologies, Seattle, WA), which included an assessment of the quality of the runs, and combined, normalized, and analyzed in nSolver and Excel.

Normalization was performed by applying a scaling factor that normalizes the geometric mean of housekeeping genes (β-actin; *ACTB*, hypoxanthine phosphoribosyltransferase 1; HPRT1 and RNA polymerase II subunit A; POLR2A) for each sample and these normalized counts were used to calculate the scores, as previously reported[92]. The median of these ISG relative expressions was used as an IFN score.

**IFN-α protein measurement by Simoa technology.** All reagents were purchased from Quanterix (reference 100860) and loaded onto the Simoa HD-1 Analyzer (Quanterix) according to the manufacturer's instructions and using three-step assay configurations. Briefly, the beads were pelleted with a magnet to remove supernatant (SN). Following several washes, 100 µL of detector antibody were added, according to the manufacturer's instructions. The beads were then pelleted with a magnet, followed by washes and 100 µL of β-D-galactosidase (SβG) were added. The beads were washed, re-suspended in

resorufin β-ᴅ-galactopy-ranoside (RGP) solution, and loaded onto the array. The array was then sealed with oil and imaged. Images of the arrays were analyzed and AEB (average enzyme per bead) values were calculated by the Simoa HD-1-associated software. Analyzer, as previously reported[93]. Human plasma samples along with calibration curves were measured using the Simoa HD-1 Analyzer. The calibration curves were fit using a 4PLfit with 1/y2 weighting factor and were used to determine the concentrations of the unknown human plasma samples. This analysis was done automatically using the software provided by Quanterix with the Simoa HD-1 Analyzer.

**IL-6, IFN-λ1, and IFN-γ protein measurements.** Concentrations were determined in patient's serum using U-PLEX Custom Human Cytokine assay (Meso Scale Diagnostics, Rockville, MD). The assays were performed according to the manufacturer's instructions with overnight incubation of the diluted samples and standards at 4 °C. The electro-chemiluminescence signals (ECL) were detected by MESO QuickPlex SQ 120 plate reader (MSD) and analyzed with Discovery Workbench Software (v4.0, MSD).

### Ex vivo stimulation of PBMCs isolated from SARS-CoV-2-infected patients and healthy donors

Bloods of SARS-CoV-2-infected patients and healthy donors were collected in EDTA tubes. PBMCs were freshly isolated by Ficoll-Hypaque density centrifugation followed by washing in pDC/PMBC culture medium (i.e., RPMI 1640 Medium supplemented with 10% FBS, 100 U/ml penicillin, 100 mg/ml streptomycin, 2 mM ʟ-glutamine, non-essential amino acids, 1 mM sodium pyruvate and 10 mM Hepes). PMBCs were frozen in 1 mL freezing medium (10% DMSO, 90% FBS) and cryopreserved in vapor phase liquid nitrogen (>−135 °C). Two hours prior to ex vivo stimulation, PBMCs were thawed out at 37 °C rinsed with 10 mL of FCS, incubated in 40 mL of pDC/PMBC culture medium at 37 C°/5% CO₂ for 30 min, and re-suspended in culture medium. 2.5 × 10 e5 PBMCs were cocultured with 1 × 10 e5 SARS-CoV-2-infected versus uninfected A549-ACE2 cells, as negative control, or were stimulated with TLR agonists [31.8 μM R848 and 42.22 μM polyI:C] for the FACS panel of mDC1/pDC analysis and LPS stimulation [2.04 μM] for the mDC2/non-mDC2/HLA-DR⁺ CD14⁺ panel in a final volume of 200 μl in 96-well round-bottom plates incubated at 37 C°/5% CO₂ for 14 to 16 h. Cell-culture SNs were collected for quantification of cytokine levels (IFN-α by ELISA/Simoa; IL-6, IFN, and IFN-λ1 by U-PLEX Custom Human Cytokine assay) while cells were harvested for flow cytometry or for Nanostring analyses.

RNAs were isolated from these cells by phenol/chloroform extraction procedure as previously described[49]. The subsequent steps of the procedure used for Nanostring analysis on ex-vivo-stimulated PBMCs were performed as described above for the patient blood samples.

### Coculture experiments using isolated pDCs

Unless indicated differently, 2 × 10 e4 pDCs were cocultured with SARS-CoV-2-infected or uninfected cells as 5 × 10 e4 or 1 × 10 e5 cells for analysis by RT-qPCR or flow cytometry, respectively, or were stimulated with 100 μl of cell-free SN collected from SARS-CoV-2-infected cells. The cells were infected at MOI 0.01, 0.1, 0.02, and 0.5 for, respectively, NCI-H358-ACE2, Huh7.5.1, A549-ACE2, and Calu-3 cells for 48 h maximum prior to collection of the cells and their SNs for coculture. As comparison pDCs were stimulated with TLR agonists [31.8 μM R848 and 42.22 μM polyI:C] in a final volume of 200 μl in 96-well round-bottom plates incubated at 37 C°/5% CO₂. When indicated, cells were cocultured in 96-well format transwell chambers (Corning) with a 0.4 μm permeable membrane. At the indicated time, cell-culture supernatants were collected for quantification of cytokine levels: IFN-α, IFN-λ1/2/3 (IL29/28 A/28B), TNF and IL-6 using a specific ELISA kit (PBL Interferon Source, Affymetrix, respectively) following the

manufacturer's instructions. Cells were harvested at the indicated times for analysis by flow cytometry or RT-qPCR.

### Immunostaining and flow cytometry analysis

At the indicated times, harvested cells were re-suspended using 2 mM EDTA-PBS solution for the coculture with PBMCs and 0.48 mM EDTA-PBS solution for pDC cocultures. Cells were incubated with 1 μL/mL viability marker diluted in PBS for 20 min at RT. After a 10-min incubation with Fc receptor blocking reagent (MACS Miltenyi Biotec) at 4 °C followed by two PBS washes, cells were stained for surface markers for 30 min at 4 °C with antibodies diluted in staining buffer (PBS without calcium and magnesium, with 2% FBS and 2 mM EDTA), followed by two PBS washes. These markers included generic lineage markers (CD3, CD19, CD20, CD56 for exclusion, and CD11c, HLA-DR for selection of cell populations), and specific markers of pDCs (CD123, BDCA-2, CD2, and Axl), mDC2 (BDCA-1), mDC1 (BDCA-3), monocytes (CD14 and CD16) and/or cell differentiation markers (CD83, CD80, and PD-L1). The references and used concentrations for the antibodies are listed in Supplementary Table 4. For the identification of apoptotic and necrotic cells, surface-stained cells were labeled using FITC Annexin V Apoptosis Detection Kit with 7-AAD according to the manufacturer's instructions. Following one wash with Annexin V Binding Buffer (Biolegend), cells were fixed with 4% PFA for 30 min at 4 °C. For intracellular immunostaining, cells were treated with 1 μl/ml GolgiPlug solution (BD Bioscience) for 3 h at 37 °C/5% CO₂ before collection. After surface staining and fixation with cytoperm/cytofix solution (BD Bioscience) for 20 min at 4 °C, IFN-α, IL-6, IFN-λ1, and TNF were stained by a 45-min incubation at 4 °C with antibodies diluted in permeabilization buffer (BD Bioscience; antibodies are listed in the Supplementary Table 4). Cells were then washed with permeabilization buffer and re-suspended in staining buffer. Flow cytometry analysis was performed using a BD LSR Fortessa 4 L using BD FACSDIVA v8.1 software. Compensation beads were used as a reference for the analysis. The data were analyzed using FlowJo 10.8.1 software (Tree Star).

### Analysis of transcriptional levels by RT-qPCR

RNAs were isolated from cells harvested in guanidinium thiocyanate citrate buffer (GTC) by phenol/chloroform extraction procedure as described previously[49]. The mRNA levels of human *MXA, ISG15, IFNL, IL6, TNF* and glyceraldehyde-3-phosphate dehydrogenase (*GADPH*) were determined by RT-qPCR using iScript RT kit (Life Technologies) and PCR Master Mix kit (Life Technologies) for qPCR and analyzed using StepOnePlus Real-Time PCR v2.3 system (Life Technologies). The sequences of the primers used for RT-qPCR are described in Supplementary Table 2. The mRNA levels were normalized to *GADPH* mRNA levels.

### Analysis of extracellular infectivity

Infectivity titers in supernatants were determined by end-point dilution in plaque assay. Briefly, 10-fold serial dilutions of SARS-CoV-2-containing supernatants were added to 2 × 10 e5 Vero cells seeded in 12-well plates for a 2 h-incubation. The medium was then replaced by DMEM containing 2% FBS and 2% carboxymethylcellulose (CMC). The cytopathic effect was scored 96 h post-infection: cells were fixed for 30 min with 4% PFA and colored by cristal violet solution.

For icSARS-CoV-2-mNG infection, foci were directly detected according to mNeongreen-positive cells. Briefly, 10-fold serial dilutions of icSARS-CoV-2-mNG-containing supernatants were added to 2 × 10 e4 Vero cells seeded in 96-well plates and fixed 24 h post-infection. GFP-expressing cells were quantified by foci counting using a Zeiss Axiovert 135 microscope.

### Viral spread assay

A549-ACE2 cells were transduced with lentiviral-based vector pseudotyped with VSV glycoprotein to stably express RFP, as previously

reported[49]. After immuno-isolation, pDCs were stained with CellTrace Violet Cell Proliferation kit (Life Technologies) for 20 min at 37 °C in the dark. Labeled pDCs were then spinned down and re-suspended in pDC culture medium. $2.5 \times 10$ e4 pDCs were cocultured with $2.5 \times 10$ e4 icSARS-CoV-2-mNG-infected cells (infected for 24 h prior to coculture) and with $2.5 \times 10$ e4 RFP⁺ uninfected cells for 48 h at 37 °C/5% $CO_2$. When indicated, the cocultures were treated with an anti-$\alpha_L$ integrin blocking antibody at 10 mg/mL. After coculture, harvested cells were stained with Live/Dead Fixable Dead Cell Stain Near-IR marker for 30 min at RT, washed with PBS and fixed with 4% PFA for 30 min at 4 °C. The level of viral spread from icSARS-CoV-2-mNG-infected cells (mNG⁺) to uninfected cells (RFP⁺) during coculture was determined by flow cytometry analysis as the frequency of infected cells (mNG⁺/RFP⁺ population) among the RFP⁺ cell population and similarly in RFP⁻ populations. Flow cytometry analysis was performed using a BD LSR Fortessa 4 L using BD FACSDIVA v8.1 software and the data were analyzed using Flow Jo 10.8.1 software (Tree Star).

### Live imaging of coculture with spinning-disk confocal microscopy analysis

A549-ACE2 cells were infected with icSARS-CoV-2-mNG for 48 h prior to coculture with pDCs. Infected cells were seeded ($2 \times 10$ e4cells per well) in a 96-Well Optical-Bottom Plate pre-coated with poly-L-lysine (1 h incubation at 37 °C/5% $CO_2$ with 8 mg/mL poly-L-lysine). Isolated pDCs were stained with 0.5 μM Vybrant cell-labeling solution (CM-DiI, Life Technologies) by successive incubations for 10 and 15 min at 37 °C and 4 °C respectively. When indicated, pDCs were incubated with anti-$\alpha_L$ integrin (10 μg/mL) 15 min prior and kept during the coculture with infected cells, when indicated, the icSARS-CoV-2-mNG-infected cells were stained with a fluorescent marker of living cells (NucSpot Live 650) by 10-min incubation at 37 °C, and 3–4 h prior to coculture with pDCs. After addition of pDCs to seeded infected cells, the cocultures were imaged every30 min for the analyses of viral replication (i.e., mNG reporter) and cell viability (i.e., live cell marker), and every 5 min for the record of anti-$\alpha_L$ integrin test with total of 24 h-record at magnification ×10 using a BSL3-based spinning-disk confocal microscope (AxioObserver Z1, Zeiss). The cocultures were maintained at 37 °C/5% $CO_2$ in an incubation chamber. Analyses of pDC motion were performed using projection of Z-stacks with maximal intensity (i.e., -5–10 selected Z-stacks per fields out of 30 Z-stacks in total). The quantification of mNeonGreen fluorescence intensity of infected cells and the duration of contacts between pDCs and infected cells were performed using Image J 1.53k Java 1.8.0_172 package (http://rsb.info.nih.gov/ij). The calculations of pDC positions were performed using Trackmate plug-in of Image J software.

### Quantitative analysis of confocal imaging

Coculture of CTV-stained-pDCs and icSARS-CoV-2-mNG-infected cells were fixed with PFA 4% at 5 or 24 h post-coculture, followed by immunostaining using anti-dsRNA and anti-Spike antibodies. Confocal imaging was performed using the LSM980 scanning confocal microscopy. For each stack of images, cocultured cells were automatically segmented based on the mNG detection, and number/frequency of dsRNA⁺ and Spike⁺ cells were quantified using a homemade Image J macro (https://github.com/jbrocardplatim/PDC-vicinity). This macro also enables us to define of the proximity of pDCs and is defined as in contact when the distance from icSARS-CoV-2-mNG-infected cells to pDCs based on pDC staining was inferiors to 5 μm.

### Gradient boosting machine learning analysis

Cell types/subsets of interest (pDCs, mDC1, mDC2, and monocytes) were exported in linear scale as csv files using the Export/Concatenate Populations FlowJow option (export CSV - Scale values with all compensated parameters - gating by the experimentalist). Each file was fingerprinted with the sample ID (patient and date of collection), cell type, acquisition cohort and activation type. This allowed straightforward characterization of each individual cells towards the expression or not of the various markers of interest using python dictionaries containing the actual gating values for each experiment. As some subpopulations of interest represent less than 1% of the gated cells, samples with <100 cells were removed from the study (commented in the python dictionaries and in accordance with all results shown in Fig. 2 and Supplementary Fig. 7). For each cell type, we generated a database where, each sample represents a line and each percentage of positive cells for a marker or percentage of these cells within the whole cell population (before gating) represents a column. Each database was studied as follows using the python library scikit-learn. First, training/test and validation sets were generated using the *train test_split function*. The obtained validation set represents 20% of the currently analyzed database. The remaining 80% samples were studied for imbalance between the different target groups (healthy, mild/asymptomatic and severe). The imbalance was handled by performing 10 random down-samplings. For each downsampling, the randomly selected samples were used to build a model with the GradientBoosting-Classifier decision tree (with n_iter_no_change set on 5)[94–96] within a 10 split cross-validate using *ShuffleSplit*, with a test set representing 20%. The efficiency of the models was measured using the accuracy scoring for each fold and for both training and test sets. All the models resulting of the cross-validation step were challenged with the validation set. The importance of each feature was then determined by permutation with the permutation_importance[97] function (with n_repeats set on 10) for the Training/Test and the Validation sets. All the accuracy scores and feature importance were stored into csv files for later graphic representation and human inspection. In order to monitor any bias in the initial train_test_split, the whole process was performed 10 times. The accuracy of the training/test sets is represented for each of the 10 iterations and the variability within the 10 subsequent down-samplings and modelings are indicated as error bars. The same representation is used for the importance of the features with the validation set (Supplementary Fig. 7). For Gradient Boosting machine learning analysis, the following software and packages were used:

1. Python 3.8.10 [GCC 9.4.0] on linux with the following packages indicated as package name [Version]: joblib [1.1.0]; matplotlib [3.5.1]; numpy [1.22.2]; pandas [1.4.0]; scikit-learn [1.0.2]

2. R (version 4.2.1): RStudio 2021.09.1 + 372; "Ghost Orchid" Release (8b9ced188245155642d024aa3630363df 611088a, 2021-11-08) for Ubuntu Bionic; Mozilla/5.0 (X11; Linux x86_64) AppleWebKit/537.36 (KHTML, like Gecko) QtWebEngine/5.12.8 Chrome/69.0.3497.128 Safari/537.36; and with the following packages indicated as package name [version]: caret [6.0.90]; ggplot2 [3.3.5]; knitr [1.37]; nnet [7.3.17]; reticulate [1.25]; tidyverse [1.3.1]; tinytex [0.36]. The code corresponding to the machine learning analysis is available in the following GitHub repository: https://github.com/dcluet/Covid_machine_learning.

### Bioinformatic analysis of the cis-acting regulatory element in the promoter

For each candidate gene, we recovered the genomic nucleotide sequence spanning from 1500 nucleotides upstream, to 100-200 nucleotides downstream of the annotated Refseq transcription start site. For genes with more than one annotated transcription start site, we collected the sequence from 1500 nucleotides upstream of the most 5′ transcription start site until 100-200 nucleotides downstream of the most 3′ transcription start site. The bioinformatics

analyses were performed using the FIMO and AME tools available on https://meme-suite.org/meme/[98,99] and the consensus sequence is as follows:

| | |
|---|---|
| NF-KB | GGGRNYYYCC |
| IRF7 | MCGAAARYGAAAVT |
| IRF3 | NSRRAAMGGAAACCGAAACYR |
| IRF1 | NTTYASTTTCACTTTCDBTTT |
| IRF5 | cCGAAACCGAAmCy |
| STAT1:STAT2 | tyAGTTTCrkTTYCy |
| IRF9 | AwCGAAACCGAAACy |

### Statistical methods

Statistical analysis was performed using R software environment for statistical computing and graphics (version 3.3.2).

For quantifications by ELISA, RT-qPCR, and flux cytometry analyses of the levels of cytokines, ISG, and cell surface markers the statistical analyses were performed using one-way ANOVA on ranks (Kruskal–Wallis rank-sum test). When the test was considered significant (p-values ≤0.05), we used the Tukey Kramer (*Nemenyi*) pairwise test as post hoc test for multiple comparisons of mean rank sums to determine which contrasts between individual experimental condition pairs were significant.

Of note, for each independent experiment preformed using PBMCs isolated from SARS-CoV-2 infected patients, the same procedures and analyses were done in parallel using different *healthy* donors, as references. Likewise, all independent experiments of cocultures with PBMCs or pDCs were performed using distinct *healthy* donors as reference. To test difference between the patient groups in the Flow cytometry analysis of the biomarker the different cell population, we used the Beta regression model with the logit link function from the R 'betareg' package, which is the suitable statistical approach for modeling continuous response Y variables that vary in the open standard unit interval (0, 1). Beta regression being unable to model in case Y contains exactly 0 or 1 values, Y data was transformed to Y', as follows:

$$Y' = (Y * (n - 1) + 0.5)/n \qquad (2)$$

where $n$ is the number of patients in all compared groups.

For the quantification by flow cytometry analysis of the viral spread of the icSARS-CoV-2-mNG SARS-CoV-2 molecular clones from RFP⁻ cells to RFP⁺ cells (initially uninfected) (Fig. 4) and for quantification of data from the live-imaging analysis using spinning-disk confocal microscopy analysis (Fig. 5), the statistical analyses were performed using one-way ANOVA followed by Tukey multiple comparisons of means. For quantification of fluorescence intensity of mNG and live cell marker, the statistical analyses were performed using pairwise comparisons using Wilcoxon rank-sum exact test and P value adjustment method: fdr. The set of Figures was prepared using PRISM software (version 8.4.3). In this study, the tests compared a significant difference between groups, and regardless whether one group is greater/smaller than another. When the statistical test gives rise to a two/one-sided test, the test herein was therefore always two-sided. In case of multiple testing, we lawfully corrected for multiple comparisons according to the analysis type. For example, when calculating pairwise comparisons between group levels with the Wilcoxon test, we used the method fdr 'false discovery rate'[100]; while in the parametric context of ANOVA, the p-values were directly adjusted by the process of Tukey multiple comparisons of means.

### Reporting summary

Further information on research design is available in the Nature Portfolio Reporting Summary linked to this article.

## Data availability

Authors confirm that all relevant data are included in the article and/or its supplementary information files, i.e., datasets used in the study along with appropriately accessible links/accession-codes. Source data are provided in this paper.

## Code availability

The code is available in the indicated GitHub repository. The code corresponding to the machine learning analysis is available in the following GitHub repository: https://github.com/dcluet/Covid_machine_learning.

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

## Acknowledgements

We thank Dr. B. Lina (CIRI, Lyon, France) for kindly providing the clinical isolate of SARS-CoV-2; Dr. Pei-Yong Shi (University of Texas Medical Branch, Galveston, US) for the infectious clone of SARS-CoV-2 with mNeonGreen reporter; Dr. V. Lotteau (CIRI, Lyon, France) for viral stock of the influenza A virus; Dr. C. Goujon (Institut de Recherche en Infectiologie de Montpellier, IRIM, France) for the ACE2-lentiviral construct and Dr. F.V. Chisari (Scripps Research Institute, La Jolla, CA) for the Huh7.5.1 cells. An illustration of Fig. 6 was created with BioRender.com. We are grateful to Drs. Y. Jaillais, A Marçais, B. Webster, S. Assil, and P.Y. Lozach for critical readings of the manuscript and to our colleagues for their encouragement and help. We acknowledge the contribution of SFR Biosciences (UMS3444/CNRS, US8/Inserm, ENS de Lyon, UCBL)

 

including the PLATIM and AniRA-cytometry facilities, especially J. Bro-card and S. Dussurgey, for technical assistance for imaging and FACS analyses, respectively. We acknowledge the contribution of the EFS Confluence/Decine-Lyon. We thank the staff members of the Occupa-tional Health and Medicine Department and of the Intensive Care Unit of Croix Rousse Hospital of the Hospices Civils de Lyon (Pr JB Fassier and Pr JC Richard, who contributed to patient recruitment and sample col-lection), the clinical research associates for theirexcellent
work and members of the clinical research and innovation department (DRCI, Hospices Civils de Lyon); and all the HCWs and patients for their participation in this clinical study. This work was supported by grants from the *Agence Nationale de la Recherche* (ANRJCJC-iSYN and ANR-22-CE15-0034_03-VERSATILE); the *Fondation pour le recherche médicale* (FRM; ANR Flash COVID-19); the *Agence Nationale pour la Recherche contre le SIDA et les Hépatites Virales* (ANRS – N21006CR and N19017CR); the *UDL/ANR IA ELAN ERC* (G19005CC); and *FINOVI* (AO11 – collaborating project); from EU H2020 ZIKAlliance. Ph.D. fellowships for M.R., G.J., and C.N. are respectively sponsored by ANRS; *'Contrats doctoraux Lyon 1 dédiés à l'International'* from Université Lyon 1 and ANRJCJC-iSYN.

## Author contributions

M.Ve., M.S.R., E.D., Ali.B., G.J., H.P., Ale.B., S.A., and M.D. for con-ception or design of the work; M.Ve., M.S.R., E.D., Ali.B., M.Vi, G.J., C.N., D.C., M.P., R.P., H.P., T.W., O.A., Ale.B., S.A., E.R., and M.D. for acquisition, analysis, or interpretation of data; D.C. and E.R. for creating new tools used in the work; M.Ve., M.S.R., E.D., Ali.B., G.J., E.R., and M.D. have drafted the initial manuscript, M.Ve., M.S.R., E.D., Ali.B., G.J., C.N., T.W., Ale.B., E.R., and M.D. for the edition of the manuscript.

## Competing interests

The authors declare no competing interests.
