## [Peer Review File · Nature Communications]

Severe COVID-19 patients have impaired plasmacytoid dendritic cell-mediated control of SARS-CoV-2-infected cellsREVIEWER COMMENTS

Reviewer #1 (Remarks to the Author):

Manuscript Background information

Venet & Ribeiro et. al. provides an analysis of plasmacytoid dendritic cell (pDCs) functions in the recognition of SARS-CoV-2, the causative agent of coronavirus disease 2019 (COVID-19). The authors clearly demonstrate that SARS-CoV-2, like many other RNA viruses and some DNA viruses, can be recognized via cell-cell contact of pDCs. Furthermore, similar to findings published by Yun et al. in *Science Immunology*, the authors indicate this cell-cell contact may promote the retention of pDC in a more IFN-I biased differentiation state. Critically, the authors confirm findings originally published in Arunachalam et al 2020, showing that pDC IFN-I production is suppressed in COVID-19 patients, and associate this phenotype specifically with severe disease. Finally, the authors provide some evidence that could suggest pDC contact directly inhibits SARS-CoV-2 replication in infected cells in vitro. Overall, the study is of great interest and timely, although the major issues listed below should be addressed.

Major Issues

1. The authors missed acknowledging a number of critical papers that either add context to or directly contradict their results. Some exclusions included:
 - a. Arunachalam et al 2020, *Science Immunology*. This study demonstrates that pDC from COVID patients show reduced IFN-I production after stimulation, and notably find no association with severity. Potential reasons that the authors findings differ from this study should be addressed.
 - b. Onodi et al 2021, *J Exp Med.*, Severa et al. 2021 *PLOS Pathogens*, and Asano et al 2021 *Science Immunology*. These studies all show that significant production of IFN-I and in some cases IFN-III after stimulation with cell free SARS-CoV-2. These are in direct contradiction of this manuscript's assertion that cell free SARS-CoV-2 does not stimulate pDC production of interferon. Notably, Onodi & Asano et al are both cited, but in neither case do the authors acknowledge that the aforementioned apparent controversy. Severa et al is especially notable because they observe IFN-I production in purified pDCs incubated with SARS-CoV-2 at an M.O.I. of 0.1 well below the M.O.I. tested by the authors. Potential reasons that the authors findings differ from these studies should be addressed.
 - c. Yun et al. 2021. *Science Immunology*. This study has already described how human pDCs may mount a distinct response after cell-cell contact. Importantly the results of this study are incredibly supportive of the authors observations. Including that pDC stimulated by cell-cell contact with infected cells retain a PD1+CD80- phenotype, as well as the IFN-I biased response (when compared to TNFa production with cell free flu). The authors of this manuscript offer novel data that this is the case in SARS-CoV-2 infection, which is important and timely, but fall short to acknowledge the similarity with data already published for other viruses. This can be easily corrected by text revisions.
2. It is interesting that the authors report no IFN-I production after stimulation with R848, a well-known agonist of TLR7 that is reported extensively to induce IFN-I production in pDCs. The authors make no mention of the unusual nature of this result. Following the path set out by Yun et al, the authors could investigate the kinetics of IFN-I induction by SARS-CoV-2 infected cells, as this is important to rule out that the differences observed are due to different kinetics of IFN production rather than the magnitude of the overall IFN-I induced by the different stimuli.
3. Related to Point #1. Flu stimulation is used as a control when analyzing PD-1/CD80 subsets of pDCs, the authors should strongly consider using Free SARS-CoV-2 to definitively show that this change in differentiation is due to the method of recognition, not the type of virus. Similar to what has already been done for Flu in Yun et al 2021.
4. The author's data concerning potential cell-cell contact mediating direct viral control is very interesting. However, the data present falls short of the amount of evidence needed to support this claim. Most importantly a reduction in NG is not necessarily the result of reduced viral replication. To give some examples of potential confounding phenomenon, this may be due to an increase in protein

turnover, a decrease in overall translation or transcription (not just viral), or a loss of cell-viability. The authors should measure viral protein and cell viability to support their important claim.

5. The gating used for IFN-I positives is mentioned in the text, but not shown (IFN α +high, vs IFN α +all). The authors indicate that the choice of gating strategy here impacts the significance of the results. All gating strategies used should be provided as supplement.

6. The conclusions from Fig 2a seem confusing as explained in the text. Some severe patients did not seem to show early IFN and this is not noted in text. It is also unclear whether the severe patients have sustained IFN-I/III at higher levels than the mild patients (as curve from mild patients is interrupted)

7. One caveat for the experiments performed in Fig 2 is that stimulation was done in non-purified pDCs and as a result the phenotypes evaluated could have been influenced by the responses of non-pDCs in the culture. This caveat should be acknowledge

Minor Issues

1. The authors should check spelling throughout the manuscript. Example Supplemental Table 1, line 19 "Positive" is misspelled.

2. The authors citations after the following statement are misleading "Studies on related coronaviruses have demonstrated that pDCs migration into the lungs, and their rapid production of IFN-I is essential to the control of lethal infections by these coronaviruses".

a. The first cited study (Lucas et al.) does not relate to pDC location in the lungs or demonstrate an essential nature for pDCs in the context of viral control. This supports neither part of the claim.

b. The second cited study (Cervantes-Barragan et al) does potentially support a role for pDC in the control of Beta-coronavirus, although the variant of mouse herpesvirus used in this case is hepatotropic, not typically appreciably measured in the lung (except in the case of IFNAR knockout or pDC depletion). This study does not measure pDC trafficking to the lung.

c. The third cited study shows pDC accumulation in the lungs of mice infected with SARS-CoV-2 but does not associate this with control.

3. The 2D projection analysis presented in Figure 2b,c are not informative and do not add clarity to the paper. Using eigen vectors generated by a dataset to make predictions about that self-same dataset is also circular. This analysis can be removed without changing the overall conclusions. If the authors wish to keep this analysis they should generate a novel data-set with new patients and see if the same predictions from these PCA plots hold for these future experiments.

4. The authors stated that "Together our results demonstrated that the monocytic subsets likely contribute to an exacerbated pro-inflammatory response implying notably IL6 production, but that is likely not triggered directly by the contact with SARS-CoV-2-infected cells." unclear why the authors suggest it is "likely not triggered directly by the contact with SARS-CoV-2-infected cells." Also note that likely is misspelled in this sentence.

5. In the representative FACS plots shown in Figure 1, the stated induction of IFN-lambda in pDCs stimulated with soluble agonist is not apparent

Reviewer #2 (Remarks to the Author):

In the manuscript "Severe Covid-19 patients have impaired plasmacytoid dendritic cell-mediated control of SARS-CoV-2-infected cells" by Venet et al., the authors investigate the activation of pDCs in Covid-10 and the response of pDCs to SARS-CoV-2 infected cells.

The manuscript is divided in two parts, in the first part the authors investigate the response of pDCs to SARS-CoV-2 infected cells and the phenotype of pDCs from COVID-19 patients, and in the second part they analyze the ability of pDCs to avoid the spreading of SARS-CoV-2.

The two parts are not related.

Major points.

Figure 1: The authors claim that pDCs are the major producers of IFN λ in response to the exposure to viral infected cells. This is not surprising. However, a comparison with conventional DCs is not shown to measure the efficiency of the response. The population named non-DCs is not

indicative, cDCs could be too few in this mixed population to measure their response. Some production of IFN/lambda1 by cDC1 shown in Figure S2 although it is difficult to compare it with pDCs. Figure 2: The PCA analysis is very confusing and does not show a true segregation of the populations. The most important point is that the number of patients analyzed, as shown in Table 1, is very low, 6 patients per group. These numbers are too low to reach any conclusion. Nevertheless, the dots in the figures (for instance in Fig.2D), which presumably correspond to the patients, are more than 6. Therefore, the way the analyses were done is confusing.

In vitro reactivation of DCs obtained from patients is not a good experiment to investigate whether DCs are functional or not, as pre-activated DCs cannot be activated again, therefore a non-response does not mean the cells are non-functional in severe COVID-19 patients, or at least it is not possible to re-stimulate the same pathways. Experiments should be controlled with pDCs obtained from patients with other types of infections. Internal control with mild early and mild late patients is difficult to interpret, since mild late patients have a reactivation response very similar to severe patients in terms of IFNalpha and IFNlambda1 production.

Figure 3: It is not clear what the point is to measure the upregulation of costimulatory molecules or the expansion of pDC subsets if the major point of the work is the investigation of the ability of pDCs to inhibit the spreading of the virus. Here again the experiment should be compared with cDCs and the ability of T cell activation of pDC should be compared with the cDC capacity. Moreover, a correlation of the expansion of pDC subsets with disease severity should be shown.

Figure 5: This is an interesting experiment but it is necessary to clarify the mechanism by which pDCs block the spread of the virus. How does integrin-mediated cell adhesion interfere with the spread of the virus?

Reviewer #3 (Remarks to the Author):

The manuscript titled "Severe COVID-19 patients have impaired plasmacytoid dendritic cell-mediated control of SARS-CoV-2-infected cells" by Venet et al investigates the role of pDC and IFNs in the control of Severe COVID. The authors utilise a range of appropriate methodologies, including patient derived samples and in vitro approaches, to demonstrate that pDC responses are altered in the setting of severe COVID, and indicate a role for direct contact via cell adhesion molecules in IFN α production and the control of infected cells. This study is topical, and of importance to the field. Some areas of this manuscript would benefit from further clarification.

Specific comments:

Do Calu-3 cells express endogenous ACE2 receptor

Figure 1a-b and line 871, figure 1 legend. Please define tPBMC. Is this total PBMC?

Page 5, Line 118, please define the soluble agonist.eg. R848+ polyI:C in text, at first use in results section

Figure 1c Vs Supp Fig 1b – Is there a discrepancy between the % of IFN lambda 1 positive pDC that are observed in Supp1b in response to SARS-CoV2 and agonist , versus the % observed in Fig 1c?

In Supp figure 1b, can the authors comment on the IFN lambda production observed in non-pDC with SARS-CoV2 but no agonist?

Page 58. There does not appear to be a reference to Supp Fig 1h in the results section.

Page 53, Fig 2g and results text Page 9, Can the authors comment on differences between CD80 and CD83 responses; severe SARS-CoV2 pDC CD80 responses appear to increase relative to healthy/ mild early/ mild late whereas agonist CD80+pDC (fig 2g) , SARS-CoV2 CD83+pDC and agonist CD83+ pDC decrease (Fig 2f)

Fig 2i and Supp Fig 2d: In its current format, the figures and results section are difficult to follow, and

would benefit from further clarification.

Do the authors have any further data as to whether integrin/ ICAM1 contact inhibits the decrease of mNG infected cell fluorescence by live cell imaging and spinning disk , as per Fig 5. Would this enable an investigation of whether the integrin/ ICAM1 inhibition changes the kinetics of this response?

Point-by-point response to the Reviewers

We thank all Reviewers for their helpful comments, which we have addressed by performing additional experiments to obtain further mechanistic insights on the activation of pDCs against SARS-COV-2 infected cells and how this is impaired in severe COVID-19 patients. The text of the manuscript was also edited according to the additional results and to address the required clarifications.

Reviewer #1 (Remarks to the Author):

Manuscript Background information

Venet & Ribeiro et. al. provides an analysis of plasmacytoid dendritic cell (pDCs) functions in the recognition of SARS-CoV-2, the causative agent of coronavirus disease 2019 (COVID-19). The authors clearly demonstrate that SARS-CoV-2, like many other RNA viruses and some DNA viruses, can be recognized via cell-cell contact of pDCs. Furthermore, similar to findings

published by Yun et al. in *Science Immunology*, the authors indicate this cell-cell contact may promote the retention of pDC in a more IFN-I biased differentiation state. Critically, the authors confirm findings originally published in Arunachalam et al 2020, showing that pDC IFN-I production is suppressed in COVID-19 patients, and associate this phenotype specifically with severe disease. Finally, the authors provide some evidence that could suggest pDC contact directly inhibits SARS-CoV-2 replication in infected cells in vitro. Overall, the study is of great interest and timely, although the major issues listed below should be addressed.

(Reply) We thank this Reviewer for pointing out the interest of our study and her/his helpful suggestions, which stimulated additional experiments.

Major Issues

1. The authors missed acknowledging a number of critical papers that either add context to or directly contradict their results. Some exclusions included:

a. Arunachalam et al 2020, *Science Immunology*. This study demonstrates that pDC from COVID patients show reduced IFN-I production after stimulation, and notably find no association with severity. Potential reasons that the authors findings differ from this study should be addressed.

b. Onodi et al 2021, *J Exp Med.*, Severa et al. 2021 *PLOS Pathogens*, and Asano et al 2021 *Science Immunology*. These studies all show that significant production of IFN-I and in some cases IFN-III after stimulation with cell free SARS-CoV-2. These are in direct contradiction of this manuscript's assertion that cell free SARS-CoV-2 does not stimulate pDC production of interferon. Notably, Onodi & Asano et al are both cited, but in neither case do the authors acknowledge that the aforementioned apparent controversy. Severa et al is especially notable because they observe IFN-I production in purified pDCs incubated with SARS-CoV-2 at an M.O.I. of 0.1 well below the M.O.I. tested by the authors. Potential reasons that the authors findings differ from these studies should be addressed.

c. Yun et al. 2021. *Science Immunology*. This study has already described how human pDCs may mount a distinct response after cell-cell contact. Importantly the results of this study are incredibly supportive of the authors observations. Including that pDC stimulated by cell-cell contact with infected cells retain a PD1+CD80- phenotype, as well as the IFN-I biased response (when compared to TNF α production with cell free flu). The authors of this manuscript offer novel data that this is the case in SARS-CoV-2 infection, which is important and timely, but fall short to acknowledge the similarity with data already published for other viruses. This can be easily corrected by text revisions.

(Reply) We thank this Reviewer for her/his suggestion to further discuss some selected publications on the topic. We have provided a more detailed discussion, including the possible explanations on the different observations brought by these complementary studies, mostly obtained using distinct setups and biological materials.

Firstly, as pointed out by this Reviewer, Arunachalam *et al* (*Science*, 2020) reported an assessment of the immune response against COVID-19 patients by a systems biology approach. Notably, they showed that pDCs from COVID-19 patients have a reduced IFN-I production upon stimulation by agonists (*i.e.*, polyIC+R848). This is in accordance with our results (**Fig. 2**). This previous study failed to highlight significant differences in pDC response across patient severity, yet it was limited to a total of 17 PBMC samples collected at heterogenous time-points post-first symptoms among patients. This is distinct from our analysis, which was performed with higher number of patients and at several time-points post-first symptom (*i.e.*, 4 or more time points) comprised within a similar time window for the two groups of severity (**Fig. 2** and **Extended Data Fig. 2**). These aspects are now better highlighted in the Discussion section (Page 15), as follows:

'Second, patients with severe diseases exhibit reduced circulating pDCs and low plasma IFN-I/ λ levels when

compared to mild/asymptomatic COVID-19 patients^{7,19,20}, yet the impact of pDC response on the severity is still elusive in other publication⁷⁴.’

All the prior reports mentioned by this Reviewer (*i.e.*, Onodi *et al.*, 2021 J Exp Med, Severa *et al.*, 2021 PLOS Pathogens and Asano *et al.*, 2021 Science Immunology) were mainly focused on the response to synthetic TLR agonists and/or SARS-CoV-2 supernatants. Whilst, we also analyzed as comparison the pDC response to cell-free SARS-CoV-2 (*i.e.*, supernatant); here our experimental approach is original, as pDCs are activated by their physical contact with infected cells. Therefore, as it stands, our analysis of the mechanism of pDC response and its impact in COVID-19 patients is completely novel and distinct from previous reports. Nonetheless in regards to soluble agonist (synthetic and SARS-CoV-2 supernatants) and in accordance with previous reports, we also showed that these types of pDC stimulation induce a potent upregulation of some activation markers (*e.g.*, HLA-DR, CD83, CD80, PD-L1), while the IFN-I/ λ response was very modestly induced as compared to the direct contact with SARS-CoV-2-infected cells. We showed here that the nature of pDC response to SARS-CoV-2 supernatants *versus* contact with infected cells is different. Importantly, the analysis of the pDC response to SARS-CoV-2 supernatants can be limited by unrelated factors, which can explain the distinct results compared to previous studies. We carefully designed our experiments to rule out these aspects, and this can explain the distinct results compared to previous studies. First, to limit possible confounding contaminations by cellular debris and/or floating cells, we selected a duration for SARS-CoV-2 infection, prior to coculture, so that no cytolytic effect was detectable when infected cells/SNs were collected for coculture. In this set-up, no or very low pDC activation by cell-free SN is detected even at a multiplicity of infection (MOI) of 5 per pDC. Secondly, the importance of cell-to-cell contacts for the transmission of the immunostimulatory signal to pDCs was directly validated in transwell chambers containing SARS-CoV-2-infected cells and pDCs separated by a 0.4 μ m permeable membrane. This physical separation of cells, but not of liquid diffusion, fully prevented IFN α production by pDCs. These distinctive aspects are now further discussed in the Results Section, as follows (Page 6):

‘We observed that cell-free SN from SARS-CoV-2-infected cell types failed to trigger IFN α production by PBMCs or by purified pDCs (**Fig. 1a-b**), even using a multiplicity of infection (MOI) of 5 per pDC. Of note, to avoid possible misinterpretation due to contamination by cell debris and/or floating cells, we selected here an incubation time for SARS-CoV-2 infection so that no cytolytic effect was detectable in infected human lung-derived cells when cells/SNs were collected for coculture. This specific set up might contribute to the different observation compared to previous reports showing a pDC IFN α production triggered by SARS-CoV-2 SN³⁴⁻³⁶. Importantly, to further determine if cell-to-cell contacts were required for the transmission of the immunostimulatory signal to pDCs, we used transwell chambers containing SARS-CoV-2-infected cells and pDCs separated by a 0.4 μ m permeable membrane. This physical cell separation fully prevented IFN α production by pDCs (**Fig. 1e-f**).’

Asano *et al.* (*X-linked recessive TLR7 deficiency in ~1% of men under 60 years old with life-threatening COVID-19*; 2021 Science Immunology) provided key genetic evidence that points out the central role of the TLR7 pathway in the severity of COVID-19. This is now included in the Introduction Section of the revised text, as follows (Page 3):

‘Accordingly, genetic deficiency (*e.g.*, X-linked recessive TLR7 deficiency), neutralization by autoantibodies directed against the IFN-I system, or viral-mediated inhibition of the IFN-I/ λ response aggravates SARS-CoV-2 pathogenesis²¹⁻²⁷.’

Lastly, the elegant recent report on pDC response to CMV-infected cells (Yun *et al.*, 2021 Science Immunology) is in line with our findings here on SARS-CoV-2. This is in accordance with our previous reports on Dengue and Chikungunya virus (Assil *et al.*, 2019 Cell Host and Microbe, Webster *et al.*, 2018 eLIFE; Décembre *et al.*, 2014 Plos Pathogens). The reference to this report by Yun *et al.* is now included in the Discussion section, as follows (Page 16):

‘We previously reported that, in the context of other viral infections (*i.e.*, various RNA genome viruses such as Dengue, Chikungunya, Hepatitis C, Zika viruses), contact with infected cells similarly induced a IRF7/IFN-I/ λ prioritized signaling in activated pDCs, as opposed to incubation of pDCs with cell-free viruses^{49,70}. More recently, similar features of pDC activation upon cell contact were also elegantly observed for a DNA genome virus, the human cytomegalovirus (CMV)⁸⁰.’

2. It is interesting that the authors report no IFN-I production after stimulation with R848, a well-known agonist of TLR7 that is reported extensively to induce IFN-I production in pDCs. The authors make no mention of the unusual nature of this result. Following the path set out by Yun *et al.*, the authors could investigate the kinetics of IFN-I induction by SARS-CoV-2 infected cells, as this is important to rule out that the differences observed are due to different kinetics of IFN production rather than the magnitude of the overall IFN-I induced by the different stimuli.

(Reply) As pointed out by this Reviewer, TLR stimulation by agonist did not trigger detectable IFN α ⁺ IFN λ 1⁺ pDCs at 14-16 hours post-stimulation, yet IFN λ 1⁺IFN α ⁻ pDCs are readily detected (**Extended Data Fig. 1b**), along with activation markers such as CD83 and PD-L1 (**Extended Data Fig. 1c-d**). All these different upregulated markers thus validated our stimulation protocols. Of note, we have now included representative dot blots for a healthy donor that demonstrated the detection of IFN λ 1⁺IFN α ⁻ pDCs upon TLR7 agonist stimulation (revised **Fig. 1c**, upper/right panel). Moreover, in accordance with our previous reports and other publications (Assil *et al.*, 2019 Cell Host and Microbes; Webster *et al.*, 2018 eLIFE; Décembre *et al.*, 2014 Plos Pathogens), we showed that secreted IFN α and IFN λ 1 can be readily detected in the supernatants from activated pDCs upon stimulation by TLR7 agonists at 16 hours post-coculture (**Fig. 4b** and **Extended Data Fig. 1h**).

As suggested by this Reviewer, we have now also analyzed the kinetics of pDC response to SARS-CoV-2 infected cells *versus* stimulation by synthetic TLR7 agonists at 4, 18 and 29 hours post-coculture. The results are presented in new panels in **Extended Data Fig. 3e** and **Extended Data Fig. 4b-c**. These results included the quantification of IFN α in the supernatant upon stimulation of isolated pDCs. This analysis showed that pDC IFN α secretion stimulated with synthetic agonists was detected as early as 4 hours post-stimulation and plateaued, and thus lower at later time points as compared to IFN α production by pDCs triggered by SARS-CoV-2-infected cells or cell-free influenza virus (new **Extended Data Fig. 4b**). In accordance, intracellular IFN α in pDCs was below detection limit when induced by synthetic agonists at any time point, as opposed to the robust levels detected upon coculture with SARS-CoV-2-infected cells and stimulation by cell-free influenza virus (new **Extended Data Fig. 4c**, left panels). In accordance with these observations, kinetic analyses of pDC diversification into PD-L1/CD80 subsets further revealed that PD-L1⁺ CD80⁺ and PD-L1⁻ CD80⁺ subsets were readily observed upon stimulation by synthetic agonist as opposed to pDC PD-L1⁺ CD80⁻ subset most exclusively detected upon contact with SARS-CoV-2-infected cells (new panel, **Extended Data Fig. 3e**). These new results are now described in the text of Result Section, as follows:

‘In contrast, the stimulation by soluble TLR agonists [R848 and polyI:C] elicited IFN λ ⁺ pDCs, but no detectable IFN α ⁺ cells (**Fig. 1c-d** and **Extended Data Fig. 1b**), yet a potent upregulation of surface expression of activation markers including CD83 and the programmed cell death ligand-1 (PD-L1) as compared to coculture with SARS-CoV-2-infected cells’ (**Extended Data Fig. 1c-d**)’ (Page 5).

‘Similar observations were made at different time points post-coculture (**Extended Data Fig. 3e**)’ (Page 11).

3. Related to Point #1. Flu stimulation is used as a control when analyzing PD-1/CD80 subsets of pDCs, the authors should strongly consider using Free SARS-CoV-2 to definitively show that this change in differentiation is due to the method of recognition, not the type of virus. Similar to what has already been done for Flu in Yun et al 2021.

(Reply) The requested experiments of pDCs stimulated with supernatants from SARS-CoV-2 infected cells are included in the analyses of PD-L1/CD80 subsets and compared to the stimulation by cell-free influenza virus in **Fig. 3d**. The impact of supernatants from SARS-CoV-2 infected cells was also tested on the expression levels of HLA-DR, CD70, PD-L1/CD83, and mTRAIL by pDCs, as well as the pDC subsets: CD2^{low}, CD2^{hi}, CD2^{hi} CD5⁻ AXL⁻ and CD2^{hi} CD5⁺ AXL⁺/AS pDC-like (**Fig. 3a-c** and **Fig. 3e** and **Extended Data Fig. 3b-d**). This experimental condition (*i.e.*, labeled as ‘SN’ in **Fig. 3**) has been further explained in the Figure Legend and Result Sections. We showed that SARS-CoV-2 supernatants induces an upregulation of some these activation markers, but not (or very poorly) IFN-I/ λ response. This pDC response is thus qualitatively distinct from the robust IFN-I/ λ response triggered by direct contact with SARS-CoV-2-infected cells.

Of note, pDC response to SARS-CoV-2 supernatants can be limited by unrelated factors (as mentioned in point #1 of this Reviewer). In this regard, we now included the results of activation marker expression upon pDCs cultured with infected cells in Transwell chambers (*i.e.*, containing SARS-CoV-2-infected cells and pDCs separated by a 0.4 μ m permeable membrane). The results of this kinetic analysis are presented in new panels in **Extended Data Fig. 3e** and **Extended Data Fig. 4b-c** and designed as [TW]. In accordance with the results obtained with SN, this physical cell separation, yet allowing liquid diffusion, prevented the production of IFN α and TNF α by pDCs at any time post-coculture (**Extended Data Fig. 4b-c**). These new results are described in the Result Section, as follows:

‘Again, supernatants from SARS-CoV-2-infected cells as well as the physical cell separation, yet allowing liquid diffusion prevented cytokine production by pDC at any time post-coculture (**Fig. 3b-e** and **Extended Data Fig. 4b-c**)’ (Page 12).

4. The author’s data concerning potential cell-cell contact mediating direct viral control is very interesting. However, the data present falls short of the amount of evidence needed to support this claim. Most importantly a reduction in NG is not necessarily the result of reduced viral replication. To give some examples of potential confounding phenomenon, this may be due to an increase in protein turnover, a decrease in overall translation or transcription (not just viral), or a loss of cell-viability. The authors should measure viral protein and cell viability to support their important claim.

(Reply) As suggested by this Reviewer, we now performed side-by-side analyses at single-cell of the levels of cell viability together with viral control and in association with the tracking of pDC contact with infected cells. The results are now included in new panels in **Fig. 5h** and **Extended Data Fig. 5f-i**. The results demonstrated that the control of viral replication is not explained by cell death of the targeted infected cells. This is in accordance with results obtained in **Fig. 3e-g** showing that:

i) mTRAIL upregulation was more limited upon pDC coculture with SARS-CoV-2-infected cells as compared to stimulation by cell-free stimulation (**Fig. 3e**)

ii) the frequencies of Annexin V⁺/7-ADD⁺ apoptotic Calu-3 and A549-ACE2 cells in cocultures with pDCs were similar between infected (*i.e.*, pDC activation) and uninfected (*i.e.*, no pDC activation) conditions (**Fig. 3f**)

iii) the contact with activated pDCs did not markedly impact the viability of the cocultured SARS-CoV-2-infected cells (*i.e.*, when comparing condition with or without the anti- α L integrin, which inhibits contact and pDC activation), nor the viability of activated pDCs themselves, even when analyzed after 48 hours of coculture (**Fig. 3g**).

These new results are described in the Result Section, as follows:

‘We further performed side-by-side analyses of the levels of cell viability (*i.e.*, live cell marker) and viral control (*i.e.*, mNG fluorescent reporter) at single-cell level and in association with the tracking of pDC contact with infected cells (**Extended Data Fig. 5f**). The results demonstrated that the control of viral replication (*i.e.*, mNG fluorescent reporter, green bars in **Fig. 5h** and **Extended Data Fig. 5g**) is not strictly explained by cell death of the targeted infected cells, as the intensity of the live cell marker was comparable when viral replication was

inhibited or not (*i.e.*, live cell marker, red bars in **Fig. 5h** and green curves in **Extended Data Fig. 5h**). The levels of live cell markers were comparable in infected cells cocultured or not with pDCs (**Fig. 5h** and **Extended Data Fig. 5h-i**), while the intensity of this live cell marker diminished upon addition of recombinant IFN β (**Fig. 5h** and **Extended Data Fig. 5j**). These results are in accordance with flow cytometry analysis of the cell-death marker expressions and living cells (**Fig. 3e-g**).’ (Page 13).

As mentioned by this Reviewer, the IFN-I/ λ response is known to induce a shut-down of cellular translation along with a modulation of the transcriptional activity. We agree that these aspects can also contribute to the control of viral infection and this is now discussed in the text of Discussion section, as follows:

‘Importantly, our results suggest that pDC response controls viral replication primarily in the infected cells in direct contact. This viral control does not seem to involve cell death of the targeted infected cells. We thus proposed that IFN-I/ λ response is concentrated at the contact site and thus potent to induce host antiviral effectors along with other regulations of host pathways expected to occur in response to IFN-I/ λ signaling, *e.g.*, translation shut-down and modulation of transcriptional activity. All of this host changes can be at play to robustly control viral replication in the targeted infected cells.’ (Page 16).

5. The gating used for IFN-I positives is mentioned in the text, but not shown (IFN α +high, vs IFN α +all). The authors indicate that the choice of gating strategy here impacts the significance of the results. All gating strategies used should be provided as supplement.

(Reply) As requested by this Reviewer, the gating strategy for IFN α ^{hi} versus IFN α ^{+all} related to the analysis of the cohort of COVID-19 patients (**Fig. 2**) is shown as a new panel in **Fig. 2d** and described in the text of Result section as follows:

‘Similar to **Fig. 1**, the frequency of IFN α producer pDCs greatly increased in response to SARS-CoV-2-infected cells in the *healthy* donor group for levels of both IFN α ^{+all} and IFN α ^{hi} (**Fig. 2c-d**, green arrows). The IFN α ^{hi} pDCs corresponds to pDCs gated when highly positive for IFN α (**Fig. 2d**).’ (Page 8).

6. The conclusions from Fig 2a seem confusing as explained in the text. Some severe patients did not seem to show early IFN and this is not noted in text. It is also unclear whether the severe patients have sustained IFN-I/III at higher levels than the mild patients (as curve from mild patients is interrupted).

(Reply) The **Fig. 2a** presents the analysis of markers in the sera of infected patients in the absence of *ex vivo* stimulation. As pointed out by this Reviewer, some patients with severe COVID-19 displayed an IFN-I/ λ response, which is nonetheless delayed compared to the group of mild/asymptomatic patients. To further characterize this difference, we have performed new analyses of these markers including additional time points post-symptom to better define the evolution over time of IFN-I/ λ response in the sera of the group of mild/asymptomatic patients. These results are now presented in **revised Fig. 2a**, especially for the detection of IFN α , IFN λ 1 and IFN γ , and demonstrate that IFN-I/ λ level diminishes over time in the mild/asymptomatic patients and are undetectable by day 40 post-symptom for all of them. These aspects are now better discussed in the Result section, as follows:

‘(..) and IFN α were undetectable by day 40 post-symptom for all the group of *Mild/asymptomatic patients* (**Fig. 2a**, second panel)’ (Page 7).

7. One caveat for the experiments performed in Fig 2 is that stimulation was done in non-purified pDCs and as a result the phenotypes evaluated could have been influenced by the responses of non-pDCs in the culture. This caveat should be acknowledged.

(Reply) We thank this Reviewer for her/his comment. Firstly, we cannot perform *ex vivo* experiments with purified pDCs from patients, since in accordance to the French legislation the maximum volume of blood samples that can be collected from patients is limited to 5-10 mL. The pDC frequency is 0.2-to-0.5% of total PBMCs, thus purification of pDCs from 10 mL is impossible. pDC isolation is usually performed from a 500mL-blood unit and allows to obtain around 10⁶ cells (*i.e.*, number of pDCs required for a single experiment). Secondly, this potential impact of non-pDCs is addressed by showing a comparable antiviral response by pDCs

from healthy donors when performed using PBMCs *versus* isolated pDCs. This was demonstrated in different types of analyses and for various parameters, see notably **Fig. 1a-d** and **Fig. 2c-h** (PBMCs) *versus* **Fig. 3, Fig. 4** and **Extended Data Fig. 3 and 4** (isolated pDCs). Thirdly, the comparison using PBMCs from patients has the advantage to provide insights into the potential relative contributions of the different hematopoietic cell types in the same samples from the patients. These aspects are now further included in the Discussion Section, as follows:

‘We demonstrated a comparable antiviral response by pDCs when performed using PBMCs *versus* isolated pDCs from healthy donors in different types of analyses and for various parameters in **Fig. 1a-d** and **Fig. 2c-h** (PBMCs) *versus* **Fig. 3, Fig. 4** and **Extended Data Fig. 3 and 4** (isolated pDCs). The comparison using PBMCs from patients has the advantageous to provide insights on the relative contributions of the different hematopoietic cell types in the same patient samples’ (Page 17).

Minor Issues

1. The authors should check spelling throughout the manuscript. Example Supplemental Table 1, line 19 “Positive” is misspelled.

(Reply) We now have checked the spelling throughout the manuscript.

2. The authors citations after the following statement are misleading “Studies on related coronaviruses have demonstrated that pDCs migration into the lungs, and their rapid production of IFN-I is essential to the control of lethal infections by these coronaviruses”.

a. The first cited study (Lucas et al.) does not relate to pDC location in the lungs or demonstrate an essential nature for pDCs in the context of viral control. This supports neither part of the claim.

b. The second cited study (Cervantes-Barragan et al) does potentially support a role for pDC in the control of Beta-coronavirus, although the variant of mouse herpesvirus used in this case is hepatotropic, not typically appreciably measured in the lung (except in the case of IFNAR knockout or pDC depletion). This study does not measure pDC trafficking to the lung.

c. The third cited study shows pDC accumulation in the lungs of mice infected with SARS-CoV-2 but does not associate this with control.

(Reply) We thanks this Reviewer for the request of clarification about the citations: the publication by Chen *et al.* (2010 J. Virol.) is the only one demonstrating the pDC migration in the lung infected by related coronaviruses, while the report by Lucas et al. illustrated that an early and transient IFN-I response is associated with moderate COVID-19 disease and the publication by Cervantes-Barragan *et al.* only showed an early control of the coronavirus MHV infection through pDC-derived IFN-I. In accordance, the previous sentence:

‘Studies on related coronaviruses have demonstrated that pDCs migration into the lungs, and their rapid production of IFN-I is essential for the control of lethal infections by these coronaviruses^{37,38,19.}’

This is now corrected as follows (Page 4):

‘Studies on related coronaviruses have demonstrated that pDCs migration into the lungs³⁷, and others also demonstrated a viral control by pDC-derived IFN-I³⁸. In accordance, reports on SARS-CoV-2 showed that an early and transient IFN-I response is associated with moderate COVID-19 disease (e.g.¹⁹).’

3. The 2D projection analysis presented in Figure 2b,c are not informative and do not add clarity to the paper. Using eigen vectors generated by a dataset to make predictions about that self-same dataset is also circular. This analysis can be removed without changing the overall conclusions. If the authors wish to keep this analysis they should generate a novel data-set with new patients and see if the same predictions from these PCA plots hold for these future experiments.

(Reply) We agree with the reviewer that the PCA analysis had some limitations. To address this issue, we have now implemented a machine learning approach based on Gradient Boosting

to monitor the relative importance of the different cell types and markers in predicting the severity/group of patients.

To this aim, we used the following approach for each cell type: due to the low number of samples, ten dataset sets (training/test: 80% and validation: 20%) were randomly generated to monitor any bias. As the training/test sets were imbalanced for the severity of the patients (healthy, mild, severe) 10 down-sampling to the lowest populated class were then performed. For each of these down-sampling a model was generated with 10 cross-validation steps for which the down-sampled training/test set was split into training set (80%) and test set (20%). The model generated for each down-sampling was then challenged with the corresponding validation set. The relative importance of each cell feature was monitored using a permutation approach on the validation set. This allowed to assess the performance of the model and to obtain the relative importance of each variable as a predictor of patient status. Results of this analysis are now shown in **Fig. 2b** (and replace the previous PCA analyses). They indicate that pDC-associated markers are the best predictors of patient status (with a prediction accuracy of 50%), followed by HLA-DR⁺ CD14⁺ and mDC1 cells (with a 48 and 47% prediction accuracy, respectively). We believe that this method pipeline could be useful for future predictive analyses. These new results are described in the Results section in Page 8, the Methods is now included along with additional information in a new **Extended Data Fig. 6**.

4. The authors stated that “Together our results demonstrated that the monocytic subsets likely contribute to an exacerbated pro-inflammatory response implying notably IL6 production, but that is likely not triggered directly by the contact with SARS-CoV-2-infected cells.” unclear why the authors suggest it is “likely not triggered directly by the contact with SARS-CoV-2-infected cells.” Also note that likely is misspelled in this sentence.

(Reply) This is now clarified in the text, as follows:

‘Together our results demonstrated that the monocytic subsets likely contribute to an exacerbated pro-inflammatory response implying notably IL6 production. The monocytic subsets do not produce IL6 in response to incubation with infected cells in our *ex vivo* coculture (**Extended Data Fig. 2c**), suggesting that IL6 production by these cells might be via indirect activation and/or happening at a different time-point’ (Page 10).

5. In the representative FACS plots shown in Figure 1, the stated induction of IFN-lambda in pDCs stimulated with soluble agonist is not apparent.

(Reply) As shown in **Extended Data Fig. 1b**, the frequency of IFNλ⁺ pDCs is quite heterogeneous among healthy donors upon stimulation of PBMCs by soluble TLR agonist. On the contrary, we observed a robust IFN-I/λ production in response to SARS-CoV-2 infected cells (**Fig. 1d**). The previous FACS plots from one donor with lower frequency of IFNλ⁺pDCs shown in **Fig. 1c** is now replaced by the FACS results from other donors, demonstrating that IFNλ⁺pDCs are detected upon stimulation by TLR agonist (revised **Fig. 1c**, upper panels).

This is now clarified in the text, as follows:

‘In contrast, the stimulation by soluble TLR agonists [R848 and polyI:C] elicited to some extent IFNλ⁺ pDCs, but no detectable IFNα⁺ cells (**Fig. 1c-d and Extended Data Fig. 1b**), yet a potent upregulation of surface expression of activation markers including CD83 and the programmed cell death ligand-1 (PD-L1) as compared to coculture with SARS-CoV-2-infected cells (**Extended Data Fig. 1c-d**)’ (Pages 5).

Reviewer #2 (Remarks to the Author):

In the manuscript “Severe COVID-19 patients have impaired plasmacytoid dendritic cell-mediated control of SARS-CoV-2-infected cells” by Venet et al., the authors investigate the activation of pDCs in COVID-19 and the response of pDCs to SARS-CoV-2 infected cells. The manuscript is divided in two parts, in the first part the authors investigate the response of pDCs to SARS-CoV-2 infected cells and the phenotype of pDCs from COVID-19 patients, and

in the second part they analyze the ability of pDCs to avoid the spreading of SARS-CoV-2. The two parts are not related.

(Reply) We thank this Reviewer for all the helpful comments. We think, as also pointed out by the Reviewer #3, that the different methodologies combined together here (*i.e.*, including patient-derived samples along with *in vitro* analysis of the molecular bases) are complementary to approach the question with both clinical and mechanical aspects. Nonetheless, the text was edited to better highlight the connection between these different levels of investigation.

Major points.

Figure 1: The authors claim that pDCs are the major producers of IFN λ in response to the exposure to viral infected cells. This is not surprising. However, a comparison with conventional DCs is not shown to measure the efficiency of the response. The population named non-DCs is not indicative, cDCs could be too few in this mixed population to measure their response. Some production of IFN λ by cDC1 shown in Figure S2 although it is difficult to compare it with pDCs.

(Reply) We thank this Reviewer for her/his suggestion. We have now analyzed the antiviral response focusing on the conventional DCs, and the mDC1 and mDC2 subsets as compared to pDCs. The conventional DCs were gated on viability, lineage⁻ [CD3, CD19, CD20, CD56, CD14, CD16], HLA-DR⁺, CD123⁻ and CD11c⁺ (*i.e.*, non-pDC enriched mDCs). Then conventional DC subsets were further sorted as CD11c⁺/BDCA3⁺ and CD11c⁺/BDCA3⁻ for the mDC1 and mDC2 subsets, respectively (**Extended Data Fig. 1a**). The IFN-I/ λ response of this different gated population was defined (revised **Fig. 1c-d**). The results demonstrated that none of the conventional DCs and subsets mount an IFN-I/ λ response upon contact with SARS-CoV-2 infected cells (revised **Fig. 1c-d**). These new results are now included in the Results section, as follows:

‘We further demonstrated that other DC subsets, referred to as non-pDC enriched mDCs, and further gated as mDC1 and mDC2 did not produced detectable IFN-I/ λ upon contact with infected cells (**Fig. 1c-d**, and see gating strategy in **Extended Data Fig. 1a**).’ (Page 5).

Figure 2: The PCA analysis is very confusing and does not show a true segregation of the populations. The most important point is that the number of patients analyzed, as shown in Table 1, is very low, 6 patients per group. These numbers are too low to reach any conclusion. Nevertheless, the dots in the figures (for instance in Fig. 2D), which presumably correspond to the patients, are more than 6. Therefore, the way the analyses were done is confusing.

(Reply) We thank this Reviewer for her/his suggestion to better describe the sample/number used for PCA analysis. Since patients were followed over time therefore having multiple measurements, we used all time points for each patient for our analysis. Importantly, the PCA is now replaced by another type of analysis in response to the comment of Reviewer #1. As above-mentioned to Reviewer #1 and briefly, this consists in a machine learning approach based on Gradient Boosting to monitor the relative importance of different cell types and markers in predicting the severity/group of patients. For this, our patient data were split multiple times randomly as *i*) an 80% training set used to build a model, with a cross-validation approach for building, for which all parameters (*i.e.*, markers and cell types) were used as variables for the predictors and *ii*) a 20% validation set (*i.e.* samples excluded from the step of model acquisition). For each random split, the obtained model was then used to predict the patient status from the validation set. This allowed to assess the performance of the model and obtain the relative importance of each variable as a predictor. Results of this analysis are shown in new **Fig. 2b** (and replace the previous PCA analyses). They indicate that pDC associated markers are the best predictors of patient status, followed by HLA-DR⁺ CD14⁺ and mDC1 cells. We believe that this method pipeline could be useful for future predictive analyses. These new

results are described in the Results section in Pages 8, the Methods is now included along with additional information in the new **Extended Data Fig. 6**.

In vitro reactivation of DCs obtained from patients is not a good experiment to investigate whether DCs are functional or not, as pre-activated DCs cannot be activated again, therefore a non-response does not mean the cells are non-functional in severe COVID-19 patients, or at least it is not possible to re-stimulate the same pathways. Experiments should be controlled with pDCs obtained from patients with other types of infections. Internal control with mild early and mild late patients is difficult to interpret, since mild late patients have a reactivation response very similar to severe patients in terms of IFN α and IFN λ 1 production. (Reply) We thank the Reviewer for his suggestion. We think that pDCs obtained from patients with other types of infection could bring confounding conclusions, because of distinct replication kinetics, tropism, viral escape mechanisms, *etc.*... Nonetheless, we herein compared severe COVID-19 *versus* mild symptoms/asymptomatic patients knowing that for these two groups: *i*) the viral loads were in the same range, *ii*) the kinetic study in individual patients is done in comparable time-window post-symptom onset (**Extended data Table 1**) and *iii*) patient samples were collected at the same location and time in the two groups of severity (*i.e.*, first wave 2020), most likely implying related circulating SARS-CoV-2 strain. Therefore, our evidence suggested that the impairment of pDCs in severely-ill patients does not strictly result from divergent viral load and/or strain. Furthermore, in *ex vivo* experiment we studied both the stimulation of the same pathway as in patients (*i.e.*, SARS-CoV-2 infected cells) as well as distinct stimulation using TLR agonist. This aspect is now further discussed in the Discussion Section, as follows:

‘Albeit not formally demonstrated, the ‘*exhausted*’ pDCs *in vivo* might not be explained by prior exposure to SARS-CoV-2 infected cells since viral loads were similar in patients of *Mild/asymptomatic* and *severe* groups, whilst the cytokine micro-environment was greatly distinct across disease severity. (Page 17).

In addition, the potential modulation of pDC responsiveness by prior exposure to SARS-CoV-2 and/or prior pDC activation was now experimentally addressed, and presented in the **Figure** enclosed to this response letter (see below). The experimental design consisted in a first stimulation of pDCs by: *i*) coculture with SARS-CoV-2 infected cells, *ii*) TLR7 agonist and IFN β , as comparison, and *iii*) no stimulation as control. pDCs were then isolated from these three different first types of culture/stimulation (by positive BDCA4-immunoselection) (**Panel a**; *schematic representation of the experimental procedure*). Each of these batches of pDCs was stimulated again by: *i*) coculture with fresh SARS-CoV-2 infected cells, *ii*) TLR7 agonist and IFN β and *iii*) no stimulation as control. The results demonstrated that we successfully established an optimized method to re-isolate pDC from a first culture. This leads to the re-isolation of over 95% CTV⁺ pDCs (demonstrated to be virtually all BDCA2⁺ pDCs, **panel b**). The contamination by infected cells from the first exposure was as minimal as approx. 0.88% (**panels b-d**).

Figure. Impact of prior activation of pDCs on their subsequent response against infected cells. **a**, Schematic representations of the experimental procedure. Isolated pDCs stained with CTV were cocultured with icSARS-CoV-2-mNG infected A549-ACE2 cells (green), uninfected cells (black), or stimulated by the imiquimod TLR7 agonist and recombinant IFN β (IMQ+IFN β , orange), for 5 hours. Then, pDCs were re-isolated from the different cocultures based on positive selection using BDCA4-magnetic beads. The re-isolated pDCs were subsequently cocultured with another batch of infected cells or stimulated by IMQ+IFN β for 16 hours. **b**, Representative dot plots of flow cytometry analysis prior to all cocultures of the isolated CTV-stained pDCs using pDC specific marker BDCA-2 (left panels) and icSARS-CoV-2-mNG infected cells with detection of mNeongreen (mNG $^+$) infected cells expression (right panels). **c**, Representative dot plots of flow cytometry analysis of cocultures of infected cells (mNG $^+$) and CTV $^+$ pDC post-first coculture and prior (left panels) versus post pDC re-isolation (right panels), including scatters representation of total cells (upper panels) and all gated cells further analysed for detection of mNG $^+$ infected cells (middle panel) and CTV $^+$ /BDCA2 $^+$ pDCs (lower panels). **d**, Graphical representations of the percentages of CTV $^+$ pDCs among all cells post-first cocultures with infected cells [inf.], uninfected cells [ctrl] or stimulated [IMQ+IFN β], and at different pDC re-isolation steps: prior re-isolation [P], re-isolated cells [+] and non-re-isolated cells [-]. **e**, Quantification of IFN α (ELISA) in the supernatants of pDCs cocultured with SARS-CoV-2 infected cells [inf.], or stimulated by IMQ+IFN β versus control uninfected cells [ctrl] for 5 hours (empty bars) and the supernatants of pDCs re-isolated from these three latter conditions and next cocultured with SARS-CoV-2 infected cells, stimulated by TLR7 agonist versus uninfected cells for an additional 16 hours (plain bars); N.D.; not detected as below the detection limit of 10 pg/mL. Results represent means \pm SD; n=2-3 independent experiments.

Importantly, the quantification of IFN α in the supernatants of the first and second cocultures revealed that pDCs were capable to mount a robust IFN α response even post-exposure to SARS-CoV-2-infected cells, and activation by TLR7 agonist/IFN β (panel e), as demonstrated by higher response compared to the second coculture with corresponding control conditions (i.e., first bars of each group). Altogether these new preliminary results along with our previous results suggests that the non-functionality of pDCs in severe COVID-19 patients cannot only

be explained by a reduced responsiveness due to a prior exposure to SARS-CoV-2 infected cells. This very important and complicated question would require additional investigation to assert these preliminary results and to define the possible impact of the microenvironment in the context of viral infection on pDC responsiveness.

Figure 3: It is not clear what the point is to measure the upregulation of costimulatory molecules or the expansion of pDC subsets if the major point of the work is the investigation of the ability of pDCs to inhibit the spreading of the virus. Here again the experiment should be compared with cDCs and the ability of T cell activation of pDCs should be compared with the cDC capacity. Moreover, a correlation of the expansion of pDC subsets with disease severity should be shown.

(Reply) As mentioned by this Reviewer, our report is primarily focused on the importance of pDC ability to inhibit viral spreading. In accordance with previous reports, pDCs are known to have specialized function to mediate an IFN-I-mediated antiviral response, whilst the capacity to launch the T cell activation is primarily assigned to other DC subsets. The determination of the upregulation of the subset markers along with the costimulatory molecules is thus envisaged, herein, as a complementary approach to further define the profile of pDC activation induced by SARS-CoV-2 infected cells. Our analyses of the parameters across disease severity and per cell type revealed that the major changes associated to disease severity were observed in the pDCs, we thus selectively focused on pDCs, rather than on other DC subsets. Nevertheless, in accordance with this comment and as discussed in another point, we have now included the comparison of IFN α /IFN λ response in the enriched myeloid DC populations *versus* pDCs (new panels in **Fig 1c-d** and revised panel in **Extended Data Fig. 1a**),

In addition, as requested by this Reviewer, we have now included the analysis of pDC subsets defined by CD80 and PD-L1 marker expression as performed in the study of the cohort of COVID-19 patients and across disease severities in new panels **Fig. 2f**, as follows (Page 8):

‘In sharp contrast, pDCs from *Severe* patients failed to be activated by SARS-CoV-2-infected cells, as revealed by the absence or low detection of IFN α , IFN λ , CD83 and CD80/PD-L1 as compared to *healthy* donors and *mild/asymptomatic* patients (**Fig. 2c-h**; red bars and arrows).’

We have also included similar analyses for other DC subsets (*i.e.*, mDC1 and mDC2 and non-mDC2 form comparison). These new analyses are now included as new panels in **Extended data Fig. 2d-f**, and described in the text of the Result Section (Page 9).

As expected, in other cell population (*i.e.*, mDC1, mDC2, non-mDC2 and HLA-DR⁺CD14⁺ populations), other markers (*i.e.*, IFN α , IL6, CD83, CD80 and PD-L1) were not readily induced by SARS-CoV-2-infected cells even in *healthy* donors (**Extended Data Fig. 2c-f**).

Figure 5: This is an interesting experiment but it is necessary to clarify the mechanism by which pDCs block the spread of the virus. How does integrin-mediated cell adhesion interfere with the spread of the virus?

(Reply) We thank this Reviewer for his/her comment. We have now performed a new side-by-side analysis at the single-cell level of the impact of the inhibition of the cell adhesion. Our results show that inhibition by anti- α_L integrin greatly reduces the duration of contact between pDCs and infected cells as compared to untreated coculture (new panels **Fig. 5f-g**) resulting in a majority of short-duration contacts (*i.e.*, shorter than 3 hours, **Fig. 5g**) and a reduced number of contacts (*data not shown*). We demonstrated that the reduction of viral replication is restricted to cells directly in contact with infected cells and after a sustained contact duration of about 8 hours (**Fig. 5e**). The reduced contact duration is expected to prevent pDC antiviral effect, and consistently the anti- α_L integrin restored an efficient viral spread even in presence of pDCs, as demonstrated by flow cytometry analysis (**Fig. 4f-g**). These new results are described in the Results section (Pages 13)

‘Our results further showed that inhibition by anti- α L integrin greatly reduces the duration of contact between pDCs and infected cells, as compared to untreated coculture (Fig. 5f-g). The inhibition of cell adhesion molecule results in a majority of short-duration contact (*i.e.*, shorter than 3 hours, Fig. 5g) and a reduced number of contact (data not shown). Therefore, in accordance with decrease of viral replication occurring upon sustained duration of contacts (Fig. 5e and Extended Data Fig. 5e), the anti- α L integrin restored an efficient viral spread even in presence of the pDCs, as demonstrated by flow cytometry analysis (Fig. 4f-g).

Reviewer #3 (Remarks to the Author):

The manuscript titled “Severe COVID-19 patients have impaired plasmacytoid dendritic cell-mediated control of SARS-CoV-2-infected cells” by Venet et al investigates the role of pDC and IFNs in the control of Severe COVID. The authors utilise a range of appropriate methodologies, including patient derived samples and in vitro approaches, to demonstrate that pDC responses are altered in the setting of severe COVID, and indicate a role for direct contact via cell adhesion molecules in IFN α production and the control of infected cells. This study is topical, and of importance to the field. Some areas of this manuscript would benefit from further clarification.

(Reply) We thank this Reviewer for sharing her/his positive opinion on the interest of our study and her/his helpful suggestions, which stimulated additional experiments.

Specific comments:

Do Calu-3 cells express endogenous ACE2 receptor

(Reply) Several previous publications have already validated the expression of endogenous ACE2 receptor in Calu-3 cells, *e.g.*,^{1,2}, along with TMPRSS2, *e.g.*,³.

- 1 Ren, X. et al. Analysis of ACE2 in polarized epithelial cells: surface expression and function as receptor for severe acute respiratory syndrome-associated coronavirus. *Journal of General Virology* 2006, 87, 1691-1695.
- 2 Tseng, C.-T. K. et al. Apical Entry and Release of Severe Acute Respiratory Syndrome-Associated Coronavirus in Polarized Calu-3 Lung Epithelial Cells. *Journal of Virology* 2005, 79, 9470-9479.
- 3 Laporte, M. et al. The SARS-CoV-2 and other human coronavirus spike proteins are fine-tuned towards temperature and proteases of the human airways. *PLoS Pathog* 2021, 17, e1009500.

This will be included in the text, as follows:

‘Calu-3 cells, which endogenously express ACE2 *e.g.*,^{1,2}, along with TMPRSS2, *e.g.*,³ *etc...*’ (Page 18)

Figure 1a-b and line 871, figure 1 legend. Please define tPBMC. Is this total PBMC?

(Reply) This is now clarified in the Figure Legend Section, as follows:

‘Quantification of IFN α in the supernatants of total PBMCs [tPBMC]’ (Page 28).

Page 5, Line 118, please define the soluble agonist eg. R848+ polyI:C in text, at first use in results section

(Reply) This is clarified in the text of Result Section as well, and as follows:

‘In contrast, the stimulation by soluble TLR agonists [R848 and polyI:C] elicited to some extent IFN λ ⁺ pDCs, but no detectable IFN α ⁺ cells’ (Page 5).

Figure 1c Vs Supp Fig 1b – Is there a discrepancy between the % of IFN lambda 1 positive pDC that are observed in Supp1b in response to SARS-CoV2 and agonist versus the % observed in Fig 1c?

(Reply) We thank this Reviewer for pointing out this aspect of our results. The percentages of IFN λ 1⁺ pDCs (including both IFN α ⁺ and IFN α pDCs) is quite heterogenous in response to stimulation by TLR agonist (*i.e.*, spread of the dots representing individual donors in Extended Data Fig. 1b). As opposed, robust detection of IFN α ⁺IFN λ 1⁺ pDCs are triggered by SARS-CoV-2 infected cells. To present more faithfully this heterogeneity of IFN λ 1 response, the previous FACS plots from one donor with lower frequency of IFN λ ⁺ pDCs is now replaced by the FACS results from other donors, demonstrating that IFN λ ⁺ pDCs are detected upon stimulation

by TLR agonist (revised **Fig. 1c**, upper panels). This is also clarified in the text in the Result Section, as follows:

‘In contrast, the stimulation by soluble TLR agonists [R848 and polyI:C] elicited to some extent $IFN\lambda^+$ pDCs, but no detectable $IFN\alpha^+$ cells (**Fig. 1c-d** and **Extended Data Fig. 1b**), yet a potent upregulation of surface expression of activation markers including CD83 and the programmed cell death ligand-1 (PD-L1) as compared to coculture with SARS-CoV-2-infected cells (**Extended Data Fig. 1c-d**)’ (Pages 5).

In Supp figure 1b, can the authors comment on the IFN lambda production observed in non-pDC with SARS-CoV2 but no agonist?

(Reply) The percentage of $IFN\lambda^+$ in non-pDC cells upon incubation with SARS-CoV-2 infected cells can be explained, at least in part, from $IFN\lambda^+$ mDC1 (as shown in a new panel in **Extended Data Fig. 2b**) and in the new analysis of $IFN\alpha^+/IFN\lambda^+$ in the enriched myeloid DC populations (Revised **Fig. 1c**) and/or as a secondary response to pDC IFN-I/ λ production by other cell types among PBMCs.

Page 58. There does not appear to be a reference to Supp Fig 1h in the results section.

(Reply) This is now mentioned in the text of result section, as follows:

‘As control, we validated that pDCs were induced in response to agonist stimulation in this device (**Extended Data Fig. 1h**)’ (Page 6).

Page 53, Fig 2g and results text Page 9, Can the authors comment on differences between CD80 and CD83 responses; severe SARS-CoV2 pDC CD80 responses appear to increase relative to healthy/ mild early/ mild late whereas agonist CD80+pDC (fig 2g), SARS-CoV2 CD83+pDC and agonist CD83+ pDC decrease (Fig 2f)

(Reply) This is now further clarified in the Result section, as follows:

‘In sharp contrast, pDCs from severe patients failed to be activated by SARS-CoV-2-infected cells, as revealed by the absence or low detection of $IFN\alpha$, $IFN\lambda$ and CD83 and CD80/PD-L1 as compared to as compared to *healthy* donors and *Mild/asymptomatic* patients (**Fig. 2c-h**; red bars and arrows).’ (Page 8)

‘The response to agonist stimulation was also greatly limited in pDCs from *Severe* patients compared to healthy donors and patients with *Mild symptoms/asymptomatic*, notably as shown by the level of activation markers (CD83, CD80 and PD-L1; **Fig. 2f-g**; red arrows).’ (Page 8).

Fig 2i and Supp Fig 2d: In its current format, the figures and results section are difficult to follow, and would benefit from further clarification.

(Reply) The revised **Fig. 2h** (previous **Fig 2i**) and **Extended Data Fig. 2h** (previous **Extended Data 2d**) and the corresponding text of Figure Legends are now improved to better clarify the results, including edition of the graphical display.

Do the authors have any further data as to whether integrin/ ICAM1 contact inhibits the decrease of mNG infected cell fluorescence by live cell imaging and spinning disk, as per Fig 5. Would this enable an investigation of whether the integrin/ ICAM1 inhibition changes the kinetics of this response?

(Reply) We thank this Reviewer for this interesting point. We have now performed a new side-by-side analysis at the single-cell level of the impact of the inhibition of the cell adhesion. As above-mentioned to Reviewer #2, our results now showed that the inhibition by anti- α_L integrin greatly reduces the duration of contact between pDCs and infected cells as compared to untreated coculture (new panels **Fig. 5f-g**) resulting in a majority of short-duration contacts (*i.e.*, shorter than 3 hours, **Fig. 5g**) and a reduced number of contacts (*data not shown*). As demonstrated the reduction of viral replication is restricted to cells directly in contact with infected cells and after a sustained contact duration of about 8 hours (**Extended data Fig. 5e**). The reduced contact duration thus results in preventing pDC antiviral effect, and consistently the anti- α_L integrin restored an efficient viral spread even in presence of pDCs as demonstrated by

flow cytometry analysis (**Fig. 4f-g**). These new results are described in the Results section (Pages 13)

'Our results further showed that inhibition by anti- α L integrin greatly reduces the duration of contact between pDCs and infected cells, as compared to untreated coculture (**Fig. 5f-g**). The inhibition of cell adhesion molecule results in a majority of short-duration contact (*i.e.*, shorter than 3 hours, **Fig. 5g**) and a reduced number of contact (data not shown). Therefore, in accordance with decrease of viral replication occurring upon sustained duration of contacts (**Fig. 5e** and **Extended Data Fig. 5e**), the anti- α L integrin restored an efficient viral spread even in presence of the pDCs, as demonstrated by flow cytometry analysis (**Fig. 4f-g**).'

REVIEWER COMMENTS

Reviewer #1 (Remarks to the Author):

The work by Vanet and Ribeiro et al. is much improved in resubmission and many of the comments previously made have been addressed. There are still concerns about the manuscript, which (with the exception of the last point) could be addressed by alterations to the text with no further experiments.

1. Many groups have previously seen IFN α by flow cytometry after stimulation with TLR7 agonists. It is unusual that the authors do not detect this at any timepoint tested. Therefore, the authors should clarify in the text that it is unusual that they are not able to detect IFN α by flow cytometry after agonist stimulation, cite studies that are able to measure this, and speculate on why their results differ from those studies.
2. Upon review of the IFN α /all gating it seems this may not be measuring real IFN α production. After stimulation cells can increase in size and autofluorescence which may be the case in the slight enlarging of the double negative population in these plots. To determine this is not the case a non-specific isotype antibody control should be presented alongside these data to demonstrate the authors are not gating on artifactual fluorescence. This could also be resolved by removing the IFN α /all analysis, and the statements about detection of basal IFN α in severe SARS-CoV-2 patients.
3. While the analysis the authors present on the relationship between pDC function and SARS-CoV-2 infection severity are very intriguing, these data are all from a total of 12 patients (6 mild/asymptomatic, 6 severe) with a skew toward more Males (4/6) in the severe group. We acknowledge that more even study design is technically challenging, and therefore do not hold this against the authors or the validity of their results. However, these results should not be treated as definitive across the diverse populations of humanity. The authors indeed acknowledge in their response to reviewers that similar numbers of patients in Aranuchalam et al 2020 did not identify a correlation between pDC function and disease severity. While the arguments presented may be valid, it should be made clear in the text that all these analyses are done with 6 patients with mild/asymptomatic infection and 6 patients with severe infection, and that the sex distribution is skewed. All statements about correlation to severity should include reference to these caveats, and that another similarly sized study did not find the same correlations.
4. The coculture analysis is one of the most exciting and intriguing pieces of this work, and the confirmation that this does not associate with increased cell death eliminates one possible confounding factor. However, this exciting claim should also be supported by very strong evidence as addition of recombinant interferon after establishment of SARS-CoV-2 infection does not seem to be effective at restricting viral growth (Thorne et al 2021, EMBO). Given the complexities inherent in these experiments, as well as the paradigm shifting implications of the claim, the highest standard of evidence should be presented to support this assertion. Unfortunately, the authors still do not quite reach this standard of evidence. As mentioned in the previous review the loss of a fluorescent reporter is not sufficient evidence of viral replication as it may not be regulated in the same manner as viral transcript and/or proteins. While translational shutoff and increased protein turnover can associate with reduced viral replication, SARS-CoV-2 can modulate host translation to support viral translation specifically (Mendez et al 2021, Cell Rep.). Therefore, as previously mentioned single cell measurement of viral transcript and/or protein (e.g. immunofluorescence microscopy) would be needed to validate the statement that viral replication is suppressed in this context.

Reviewer #2 (Remarks to the Author):

The authors answered my concerns

Reviewer #3 (Remarks to the Author):

The authors have addressed the comments with significant additional experimental research, and in text . The manuscript is of general interest and timely for the field. Minor comment below.

Minor comment:

Line 171. The response does not appear to be conserved in all severe patients. I recommend modification to "The IFN-1/L responses was elevated in SOME severe patients."

Point-by-point response to the Reviewers

We thank all Reviewers for their helpful comments, which we have addressed by performing additional experiments to obtain further strength of the conclusions. The text of the manuscript was also edited according to the additional results and to address the required clarifications.

Reviewer #1 (Remarks to the Author):

The work by Venet and Ribeiro et al. is much improved in resubmission and many of the comments previously made have been addressed. There are still concerns about the manuscript, which (with the exception of the last point) could be addressed by alterations to the text with no further experiments.

(Reply) We thank this Reviewer for pointing out the improvement of our manuscript and her/his helpful suggestions that we have addressed as described below.

1. Many groups have previously seen IFN α by flow cytometry after stimulation with TLR7 agonists. It is unusual that the authors do not detect this at any timepoint tested. Therefore, the authors should clarify in the text that it is unusual that they are not able to detect IFN α by flow cytometry after agonist stimulation, cite studies that are able to measure this, and speculate on why their results differ from those studies.

(Reply) We thank this Reviewer for her/his comment, and edited the text accordingly, as follows:

'While the synthetic agonists induced IFN α secretion by pDCs detected by ELISA as early as 4 hours post-stimulation, yet this response is greatly lower at later time points as compared to pDC IFN α production triggered by SARS-CoV-2-infected cells (Extended Data Fig. 4b, right panels). This might explain why in our experimental setting (i.e., unexpectedly and distinct from some other studies, e.g.,^{33, 34}), the pDC IFN α production induced by synthetic agonists was below the detection limit by flow cytometry, as opposed to the robust response to SARS-CoV-2-infected cells or cell-free influenza virus (Extended Data Fig. 4c, left panels). In sharp contrast, and as validation of our experimental setting, synthetic agonists triggered a potent pDC TNF α production, markedly higher compared to pDCs cocultured with infected cells (Extended Data Fig. 4c). (Page 12).'

2. Upon review of the IFN α hi/all gating it seems this may not be measuring real IFN α production. After stimulation cells can increase in size and autofluorescence which may be the case in the slight enlarging of the double negative population in these plots. To determine this is not the case a non-specific isotype antibody control should be presented alongside these data to

demonstrate the authors are not gating on artifactual fluorescence. This could also be resolved by removing the IFN α ^{hi}/all analysis, and the statements about detection of basal IFN α in severe SARS-CoV-2 patients.

(Reply) We have performed additional experiments to better assess the specificity of IFN α detection by Flow cytometry methods. The new experiments were performed using similar detection methodology (*i.e.*, sample preparation protocol, gating strategy and antibody panels) and included a side-by-side comparison for the same PBMC samples of IFN α detection *versus* the control isotype and the omission of only this antibody, but keeping of all the other antibodies of the staining panel, and using similar Flow cytometry settings. These new results are now presented in **Extended data Fig. 2b** and thus further improve and support our conclusions. The text was edited accordingly, as follows:

*'The detection IFN α was further validated by controls, including isotype control and omission of only anti-IFN α within the same panel of antibodies *i.e.*, keeping of all the other antibodies of the staining panel, and using similar Flow cytometry settings (Extended Data Fig. 2b).' (Page 8)*

In addition, we also validated that the separation as IFN α ^{hi} and IFN α ^{all+} has no marked impact on the prediction accuracy of the analysis *via* gradient boosting machine learning method as shown in new **Fig. 2b** and addition of the panel e in **Extended data Fig. 7**. The text was edited as follows:

'Of note, a distinction for IFN α ^{hi} and IFN α ^{all+} cells led to similar predictive accuracy (Extended Data Fig. 7e).' (Page 8)

3. While the analysis the authors present on the relationship between pDC function and SARS-CoV-2 infection severity are very intriguing, these data are all from a total of 12 patients (6 mild/asymptomatic, 6 severe) with a skew toward more Males (4/6) in the severe group. We acknowledge that more even study design is technically challenging, and therefore do not hold this against the authors or the validity of their results. However, these results should not be treated as definitive across the diverse populations of humanity. The authors indeed acknowledge in their response to reviewers that similar numbers of patients in Aranuchalam et al 2020 did not identify a correlation between pDC function and disease severity. While the arguments presented may be valid, it should be made clear in the text that all these analyses are done with 6 patients with mild/asymptomatic infection and 6 patients with severe infection, and that the sex distribution is skewed. All statements about correlation to severity should include reference to these caveats, and that another similarly sized study did not find the same correlations.

(Reply) We thank this Reviewer for her/his comment, and clarified this aspect in the text accordingly, and as follows:

'Nonetheless, owing to the technical challenge to perform these functional analyses for a larger cohort of patients, future investigations including a larger diversity of groups (e.g., additional patients with anti-IFN antibody, with immunosuppressive treatments, children etc...) will enable to reach definitive conclusion across diverse human populations.' (Page 15)

4. The coculture analysis is one of the most exciting and intriguing pieces of this work, and the confirmation that this does not associate with increased cell death eliminates one possible confounding factor. However, this exciting claim should also be supported by very strong evidence as addition of recombinant interferon after establishment of SARS-CoV-2 infection does not seem to be effective at restricting viral growth (Thorne et al 2021, EMBO). Given the

complexities inherent in these experiments, as well as the paradigm shifting implications of the claim, the highest standard of evidence should be presented to support this assertion. Unfortunately, the authors still do not quite reach this standard of evidence. As mentioned in the previous review the loss of a fluorescent reporter is not sufficient evidence of viral replication as it may not be regulated in the same manner as viral transcript and/or proteins. While translational shutoff and increased protein turnover can associate with reduced viral replication, SARS-CoV-2 can modulate host translation to support viral translation specifically (Mendez et al 2021, Cell Rep.). Therefore, as previously mentioned single cell measurement of viral transcript and/or protein (e.g. immunofluorescence microscopy) would be needed to validate the statement that viral replication is suppressed in this context.

(Reply) We thank this Reviewer for her/his advice. As suggested, to directly assess that mNG reporter reflect the level of viral replication, we have now performed a combined immunofluorescence detection of other parameters of replication level, analyzed by both Flow cytometry and Confocal microscopy in kinetic studies at single-cell level and using an experimental setting as previously *i.e.*, cells infected by the recombinant SARS-CoV-2 infectious clone [icSARS-CoV-2-mNG] expressing the mNG reporter and cocultured with pDCs *versus* in the absence pDC. We detected both dsRNA – reflecting the replication intermediate species – and the Spike protein in cells defined as mNG⁺ (**Extended Data Fig 6**). Of note, virtually all mNG⁻ cells were also dsRNA⁻ and/or Spike⁻. This was further confirmed when focusing on infected cells nearby pDCs (*i.e.*, at cell-to-cell distance < 5 μm) by using Confocal microscopy imaging analyzed by automated quantification methods. This is presented in a new figure (**Extended Data Fig. 6d-f**). These new results are now described in *Figure Legend section* and in the text of results section, as follows:

'To assess that the mNG reporter reflects the replication level, we combined it with the detection of other viral replication parameters i.e., the dsRNA reflecting the replication intermediate species and the Spike protein analyzed by both Flow cytometry and Confocal imaging analysis. The results demonstrated that mNG⁺ cells also express dsRNA and/or Spike protein (Extended data Fig. 6). This was observed for the majority mNG⁺ cells by confocal analysis (Extended data Fig. 6a-c) and even detected for virtually all mNG⁺ cells when assessed by confocal imaging analysis (Extended data Fig. 6d-g). As opposed, mNG⁻ cells were also dsRNA⁻ and/or Spike⁻. These observations were further confirmed when focusing the quantification to icSARS-CoV-2-mNG-infected cells nearby pDCs [contact] as defined for pDC/infected cell distance inferior to 5 μm (Extended data Fig. 6d-f). These results demonstrated that mNG reporter reflects the replication level.' (Page 13)

Reviewer #2 (Remarks to the Author):

The authors answered my concerns

(Reply) We thank this Reviewer for her/his approval.

Reviewer #3 (Remarks to the Author):

The authors have addressed the comments with significant additional experimental research, and in text. The manuscript is of general interest and timely for the field. Minor comment below.

(Reply) We thank this Reviewer for her/his appreciative comment.

Minor comment:

Line 171. The response does not appear to be conserved in all severe patients. I recommend modification to “The IFN-1/L responses was elevated in SOME severe patients.”

(Reply) We thank this Reviewer for her/his comment and corrected the text accordingly (Page 7).

REVIEWERS' COMMENTS

Reviewer #1 (Remarks to the Author):

The authors have satisfactorily addressed all my concerns